# Node Embeddings via Neighbor Embeddings

**Jan Niklas Böhm** [+]
*Hertie AI, University of Tübingen, Germany*
mail@jnboehm.com

**Marius Keute** [+]
*Hertie AI, University of Tübingen, Germany*

**Alica Guzmán**
*Hertie AI, University of Tübingen, Germany*

**Sebastian Damrich**
*Hertie AI, University of Tübingen, Germany*
sebastian.damrich@uni-tuebingen.de

**Andrew Draganov**
*Department of Computer Science, Aarhus University, Denmark*
draganovandrew@gmail.com

**Dmitry Kobak**
*Hertie AI, University of Tübingen, Germany*
dmitry.kobak@uni-tuebingen.de

**Reviewed on OpenReview:** *https://openreview.net/forum?id=8APIU9cauZ*

[+] *Equal contribution*

## Abstract

Node embeddings are a paradigm in non-parametric graph representation learning, where graph nodes are embedded into a given vector space to enable downstream processing. State-of-the-art node-embedding algorithms, such as DeepWalk and node2vec, are based on random-walk notions of node similarity and on contrastive learning. In this work, we introduce the graph neighbor-embedding (graph NE) framework that directly pulls together embedding vectors of adjacent nodes without relying on any random walks. We show that graph NE strongly outperforms state-of-the-art node-embedding algorithms in terms of local structure preservation. Furthermore, we apply graph NE to the 2D node-embedding problem, obtaining graph $t$-SNE layouts that also outperform existing graph-layout algorithms.

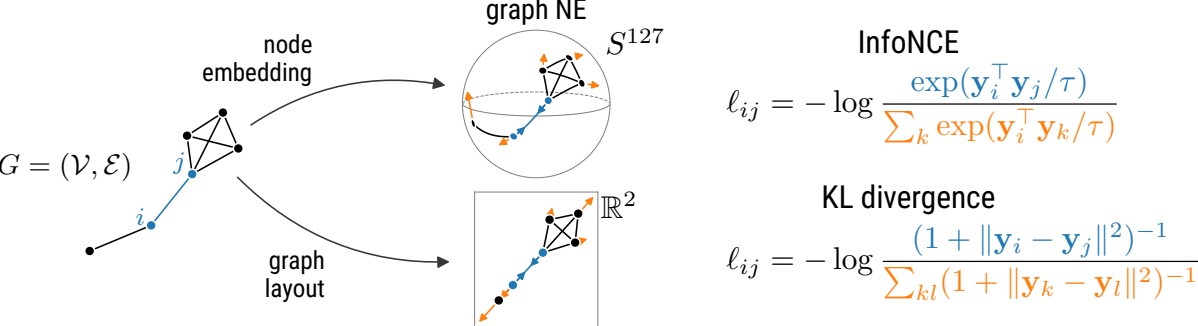

Figure 1: Graph $G = (\mathcal{V}, \mathcal{E})$ embedded into $S^{127}$ and $\mathbb{R}^2$ with graph NE. Blue denotes the attractive force between neighboring nodes $i$ and $j$ with $(i, j) \in \mathcal{E}$, orange corresponds to repulsive forces between all points.

# 1 Introduction

Many real-world datasets, ranging from molecule structures to citation networks, come in the form of graphs. A graph $G$ is an abstract object consisting of a set of nodes $\mathcal{V}$ and a set of edges $\mathcal{E}$ between them; the nodes do not inherently belong to any specific metric space. Therefore, the field of graph representation learning has emerged with the goal of embedding the nodes into a metric space, such as $\mathbb{R}^d$, so that the graph structure (neighborhoods, graph distances, etc.) is well-preserved. In this paper, we only consider *non-parametric* approaches that do not use any node features.

Popular *node-embedding* methods like DeepWalk (Perozzi et al., 2014) and node2vec (Grover & Leskovec, 2016) are based on contrastive learning and random-walk notions of node similarity, reducing the node-embedding problem to a word-embedding problem and then relying on the word2vec algorithm (Mikolov et al., 2013) for optimization. At the same time, node embeddings into $\mathbb{R}^2$ for visualization purposes — known as *graph layouts* — are typically obtained by algorithms that simply pull together neighboring (i.e. connected by an edge) nodes, traditionally using spring models (Fruchterman & Reingold, 1991). In the context of dimensionality reduction, the idea of pulling neighbors together has become known as *neighbor embeddings* through methods like *t*-SNE (van der Maaten & Hinton, 2008) and UMAP (McInnes et al., 2018). This raises the question: *Can such neighbor-embedding approaches be used for generic node-embedding problems?*

In this work, we show that neighbor-embedding methods are remarkably effective for node-embedding problems and introduce a framework called *graph neighbor embeddings (graph NE)* (Figure 1). Our work builds on recent literature which allows to effectively optimize neighbor embeddings in high-dimensional embedding spaces (McInnes et al., 2018; Damrich et al., 2023). We show that graph NE outperforms DeepWalk and node2vec in terms of local structure preservation, while being conceptually simpler (no random walks are needed) and without requiring costly hyperparameter tuning. Furthermore, we show that graph NE can also be applied for 2D node embeddings (Figure 1), outperforming existing graph-layout methods. In short, our results demonstrate that neighbor embeddings are a powerful approach to graph representation learning that beats state-of-the-art node-embedding algorithms.

# 2 Related work

**Non-parametric node embeddings**  The popular DeepWalk (Perozzi et al., 2014) and node2vec (Grover & Leskovec, 2016) algorithms optimize node placement in a high-dimensional target space based on random walks over a graph. These walks treat nodes as analogous to words and random-walk paths as sentences, enabling the application of word-embedding techniques to learn the representation. Specifically, DeepWalk achieves this by performing random walks from each starting node and then using the word2vec algorithm (Mikolov et al., 2013) to ensure that nodes which often co-occur in these random walks are represented near one another in the embedding space. The node2vec algorithm similarly obtains node embeddings by giving graph traversals to the word2vec algorithm, but it differs from DeepWalk by defining two parameters which control the depth-first vs. breadth-first nature of the random walk. These parameters ($p$ and $q$) provide an additional level of control over the community structure uncovered by the walks, with DeepWalk being a specific instantiation of node2vec when these parameters are both set to 1.

Both DeepWalk and node2vec have been widely adopted for graph-based machine learning applications, including classification and link-prediction tasks (Khosla et al., 2019). Although connections have been drawn between word2vec and contrastive learning (Saunshi et al., 2019), we emphasize that the DeepWalk and node2vec algorithms are often regarded as separate from standard contrastive techniques (Grohe, 2020).

Instead of optimizing the embedding coordinates freely, Alvarez-Gonzalez et al. (2023) found improved performance after constraining the embedding coordinates in DeepWalk based on the graph connectivity structure. Their approach, called IGEL, allows to embed new, previously unobserved nodes, which standard non-parametric methods (including ours) do not allow. Their approach can in principle be combined with any gradient-descent-based node-embedding method, including ours.

**Parametric node embeddings and node-level graph contrastive learning**  Our paper is about *non-parametric* embeddings that only use the structure of the graph $G = (\mathcal{V}, \mathcal{E})$. In contrast, *parametric*

graph contrastive learning (GCL) methods use node feature vectors and employ a neural network, usually a graph convolutional network (GCN; Kipf & Welling, 2017), to transform features into embedding vectors.

The basic principle behind contrastive learning is to learn a data representation by contrasting pairs of observations that are similar to each other (positive pairs) with those that are dissimilar to each other (negative pairs). In computer vision, positive pairs are generated via data augmentation, e.g. in SimCLR (Chen et al., 2020). GCL can be graph-level or node-level, depending on whether representations are obtained for a set of graphs or for the set of nodes of a single graph. Many graph-level (e.g. You et al., 2020) and node-level GCL algorithms (Velickovic et al., 2019; Zhu et al., 2020b; Hassani & Khasahmadi, 2020; Thakoor et al., 2022; Zhang et al., 2021; Zhu et al., 2021) are also based on graph augmentations, such as node dropping or edge perturbation. A general problem with domain-agnostic graph augmentations is that they can have unpredictable effects on graph semantics (Trivedi et al., 2022). This motivated development of augmentation-free node-level GCL methods, where positive pairs are pairs of nodes that are located close to each other in terms of graph distance (Lee et al., 2022; Li et al., 2023; Zhang et al., 2022). Recent work argued that GCL methods effectively pull connected nodes together, sometimes explicitly through their loss function, but also implicitly through the GCN architecture (Trivedi et al., 2022; Wang et al., 2023; Guo et al., 2023). A GCN can also optimize a neighbor-embedding loss on node features and/or on shortest-path distances (Leow et al., 2019).

Note that all algorithms mentioned in this paragraph are parametric and fundamentally depend on node features. In contrast, our proposed graph NE algorithm is non-parametric and operates exclusively on graph structure without requiring node features. Throughout this paper, we therefore restrict our comparisons to other non-parametric methods.

**Graph layouts**  Graph-layout algorithms have traditionally been based on spring models, where every connected pair of nodes feels a distance-dependent attractive force $F_a$ and all pairs of nodes feel a distance-dependent repulsive force $F_r$ (*force-directed graph layouts*). Many algorithms can be written as $F_a = d_{ij}^a$ and $F_r = d_{ij}^r$ (Noack, 2007), where $d_{ij}^a$ (resp. $d_{ij}^r$) is the embedding distance between nodes $i$ and $j$ raised to the $a$-th (resp. $r$-th) power. For example, the Fruchterman–Reingold algorithm uses $a = 2, r = -1$ (Fruchterman & Reingold, 1991); ForceAtlas2 uses $a = 1, r = -1$ (Jacomy et al., 2014); LinLog uses $a = 0, r = -1$ (Noack, 2007). Efficient implementations can be based on Barnes–Hut approximation of the repulsive forces, as in SFDP (Hu, 2005). ForceAtlas2 has been shown to be related to neighbor embeddings (Böhm et al., 2022).

Several recent graph-layout algorithms have been inspired by neighbor embeddings, and in particular by $t$-SNE (van der Maaten & Hinton, 2008). tsNET (Kruiger et al., 2017) applied a modified version of $t$-SNE to the pairwise shortest-path distances between all nodes. DRGraph (Zhu et al., 2020a) accelerated tsNET by using negative sampling (Mikolov et al., 2013). $t$-FDP (Zhong et al., 2023) suggested custom $F_a$ and $F_r$ forces inspired by $t$-SNE and adopted the interpolation-based approximation of Linderman et al. (2019). SGtSNEpi (Pitsianis et al., 2019) is the closest method to the 2D version of our proposed graph NE algorithm. It applies $t$-SNE optimization to affinities derived from the graph $G$, but derives these affinities in a more complex way than we do, and with additional hyperparameters (Section 4.3).

There is a separate set of methods which produce graph embeddings via classical dimensionality reduction techniques. Some of these, such as Laplacian Eigenmaps (Belkin & Niyogi, 2003) and Diffusion Maps (Coifman & Lafon, 2006), can be applied directly to graphs (and amount to eigendecomposition of the graph Laplacian). We use Laplacian Eigenmaps in our comparisons as a representative algorithm from this family. Other approaches employ variants of multidimensional scaling on graph-derived distances (Gansner et al., 2012; Miller et al., 2023; Zhang et al., 2023).

## 3 Background: Neighbor-embedding framework

### 3.1 Neighbor embeddings

Neighbor embeddings are a family of dimensionality-reduction methods aiming to embed $n$ observations from some high-dimensional metric space $\mathcal{X}$ into a lower-dimensional (usually two-dimensional) Euclidean space

$\mathbb{R}^d$, such that neighborhood relationships between observations are preserved in the embedding space. We denote the embedding vectors as $\mathbf{y}_i \in \mathbb{R}^d$.

One of the most popular neighbor embedding methods, $t$-distributed stochastic neighbor embedding ($t$-SNE; van der Maaten & Hinton, 2008), is an extension of the earlier SNE (Hinton & Roweis, 2002). $t$-SNE minimizes the Kullback–Leibler divergence between the high-dimensional and low-dimensional *affinities* $p_{ij}$ and $q_{ij}$:

$$\mathcal{L} = \sum_{ij} p_{ij} \log \frac{p_{ij}}{q_{ij}} = \text{const} - \sum_{ij} p_{ij} \log q_{ij}. \tag{1}$$

Both affinity matrices are defined to be symmetric, positive, and to sum to 1. The high-dimensional affinities $\mathbf{P}$ are computed using adaptive Gaussian kernels whose mass is concentrated on nearest neighbors. Low-dimensional affinities $\mathbf{Q}$ are defined in $t$-SNE using a $t$-distribution kernel with one degree of freedom, also known as the Cauchy kernel:

$$q_{ij} = \frac{(1 + \|\mathbf{y}_i - \mathbf{y}_j\|^2)^{-1}}{\sum_{k \neq l}(1 + \|\mathbf{y}_l - \mathbf{y}_k\|^2)^{-1}}. \tag{2}$$

In practice, $t$-SNE optimization can be accelerated by an approximation of the repulsive force field based on the Barnes–Hut algorithm (van der Maaten, 2014; Yang et al., 2013), on interpolation (Linderman et al., 2019), or on sampling (Artemenkov & Panov, 2020; Damrich et al., 2023; Draganov et al., 2023; Yang et al., 2023).

## 3.2 Contrastive neighbor embeddings

The contrastive neighbor-embedding (CNE) algorithm (Damrich et al., 2023) is a flexible dimensionality-reduction framework that replaces $t$-SNE's Kullback–Leibler divergence loss with contrastive losses, such as the InfoNCE loss (Jozefowicz et al., 2016; Oord et al., 2018). This loss function is called *contrastive* because it is based on contrasting pairs of $k$-nearest neighbors and non-neighbors in the same mini-batch, and does not require a global normalization like in Equation (2). As a result, the runtime of CNE scales like $\mathcal{O}(nd)$ with the number of points $n$ and the embedding dimensionality $d$, unlike other existing $t$-SNE implementations that scale like $\mathcal{O}(n^2d)$ (van der Maaten & Hinton, 2008) or $\mathcal{O}(n2^d)$ (Linderman et al., 2019). This enables CNE to optimize high-dimensional outputs (large $d$). CNE with the InfoNCE loss approximates $t$-SNE (Damrich et al., 2023; Ma & Collins, 2018; see also Section 4.4).

The InfoNCE loss is defined for one pair of $k$-nearest neighbors $ij$ (*positive pair*) with affinity $p_{ij}$ as

$$\ell(i, j) = -p_{ij} \log \frac{w_{ij}}{w_{ij} + \sum_{k=1}^{m} w_{ik}}, \tag{3}$$

where $w_{ij}$ are *non-normalized* low-dimensional affinities standing in for the normalized affinities $q_{ij}$ above. The sum in the denominator is over $m$ *negative pairs* $ik$ where $k$ can be drawn from all points in the same mini-batch apart from $i$ and $j$. One mini-batch consists of $b$ pairs of neighbors, and hence contains $2b$ points. Therefore, for a given batch size $b$, the maximal value of $m$ is $2b - 2$. The larger the number of negative samples $m$, the better is the approximation to $t$-SNE (Damrich et al., 2023). The InfoNCE loss aims to make $w_{ij}$ large, i.e. place embeddings $\mathbf{y}_i$ and $\mathbf{y}_j$ nearby, if $ij$ is a positive pair, and small if it is a negative one.

The $w_{ij}$ affinities do not need to be normalized. When embedding into $\mathbb{R}^2$, they can just be defined as

$$w_{ij} = (1 + \|\mathbf{y}_i - \mathbf{y}_j\|^2)^{-1}. \tag{4}$$

When using a high-dimensional embedding space, e.g. $d = 128$ instead of $d = 2$, embedding vectors are usually projected to lie on the unit sphere. For points on the unit sphere, the cosine distance and the squared Euclidean distance differ only by a constant multiplicative factor, making the following definitions of $w_{ij}$ equivalent:

$$w_{ij} = \exp\left(\frac{\mathbf{y}_i^\top \mathbf{y}_j}{\|\mathbf{y}_i\| \cdot \|\mathbf{y}_j\| \cdot \tau}\right) = \text{const} \cdot \exp\left(-\left\|\frac{\mathbf{y}_i}{\|\mathbf{y}_i\|} - \frac{\mathbf{y}_j}{\|\mathbf{y}_j\|}\right\|^2 \Big/ (2\tau)\right), \tag{5}$$

where $\tau$ is called the *temperature* (by default, $\tau = 0.5$). Together with Equation (3), this gives the same loss function as in SimCLR (Chen et al., 2020), a popular contrastive learning algorithm in computer vision. Note that instead of nearest neighbors, SimCLR uses pairs of augmented images as positive pairs.

## 4 Graph NE: Applying the neighbor-embedding framework to graphs

### 4.1 General approach

Neighbor-embedding algorithms employ high-dimensional affinities with most $p_{ij} \approx 0$. This can be seen as a generalization of discrete nearest neighbors: if $p_{ij}$ is close to 0, then the points are effectively dissimilar. However, almost the same visualizations can be obtained using hard nearest neighbors, i.e. simply by normalizing the symmetric $k$NN graph adjacency matrix $\mathbf{A}$ directly (Artemenkov & Panov, 2020; Damrich et al., 2023):

$$\mathbf{P} = \mathbf{A} \Big/ \sum_{ij} A_{ij}. \tag{6}$$

Here, $\mathbf{A}$ has element $A_{ij} = 1$ if $\mathbf{x}_j$ is within the $k$ nearest neighbors of $\mathbf{x}_i$ or vice versa. This is equivalent to simply leaving out $p_{ij}$ from Equation (3).

Thus, even though neighbor embeddings are usually not presented as such, they can be thought of as node-embedding algorithms, specifically applied to $k$NN graphs. During optimization, neighboring nodes (sharing a $k$NN edge) feel attraction, whereas all nodes feel repulsion, arising through the normalization in Equations (2) and (3).

This suggests a simple strategy, which we call *graph neighbor embedding (graph NE)*, for applying the neighbor embedding framework to a general graph $G$: obtain affinities directly from $G$ instead of a $k$NN graph of some data, and then compute a non-parametric neighbor embedding on these affinities (Figure 1).

### 4.2 High-dimensional node embeddings via graph NE

Given an unweighted graph $G = (\mathcal{V}, \mathcal{E})$, its adjacency matrix $\mathbf{A}$ has elements $A_{ij} = 1$ if $(i, j) \in \mathcal{E}$ and $A_{ij} = 0$ otherwise. Since all graphs considered in this study are undirected, the adjacency matrix is a binary, symmetric square $n \times n$ matrix. In order to convert the adjacency matrix into an affinity matrix suitable for neighbor embedding, we followed the simple normalization strategy in Equation (6). Then, graph NE optimizes the embedding using the contrastive InfoNCE loss function (through the CNE backend) to place neighbors close to each other in the embedding (Section 3.2). We used the `cne` library (Damrich et al., 2023).

For all experiments with CNE we used the output dimensionality $d = 128$, following the DeepWalk paper, and the cosine distance (meaning the embedding vectors lie on a hypersphere, Equation 5). We set the batch size to $\min\{8192, |\mathcal{V}|/10\}$ (smaller graphs required smaller batch sizes for good convergence) and used full-batch repulsion ($m = 2b - 2$) for better local structure preservation (Damrich et al., 2023). The number of epochs was set to 100. We used the Adam optimizer (Kingma & Ba, 2015) with learning rate 0.001. Graph NE was initialized with 128-dimensional Diffusion Maps (Coifman & Lafon, 2006), although we saw almost no difference when using random initialization (Figure S4b,e). For this paper, we implemented in `cne` version 0.4.0 some of the API options shown in the code snippet below.

```python
from cne import CNE
C = CNE(
    loss_mode="infonce", temperature=0.05, parametric=False, embd_dim=128,
    metric="cosine", batch_size=8192, negative_samples="full-batch", optimizer="adam"
)
Y = C.fit_transform(graph=A, init="diffmaps")
```

Note that our method is conceptually much simpler than DeepWalk and node2vec. In both of these algorithms, random walks are used to implicitly estimate node similarity by their co-occurence, and then word2vec is employed to train the embedding. Furthermore, node2vec requires per-graph hyperparameter tuning so that its random-walk distribution appropriately models the input graph (Grover & Leskovec, 2016). In our graph NE method, all nodes connected by an edge attract each other, requiring no random walks.

### 4.3 Graph layouts via 2D graph NE

A graph layout is a 2-dimensional node embedding. Therefore, we can apply graph NE in 2D to obtain graph layouts. Since the main purpose for graph layouts is visualization, we use the Cauchy similarity (Equation 2). The embedding dimensionality $d = 2$ allows us to use the KL divergence and `openTSNE` library (Poličar et al., 2024) with default parameters for optimization. It supports Diffusion Maps for initialization (Kobak & Linderman, 2021), sets the learning rate to $n$ to achieve good convergence (Linderman & Steinerberger, 2019; Belkina et al., 2019), and employs the FIt-SNE algorithm (Linderman et al., 2019). For this paper, we made some improvements to the spectral initialization in `openTSNE`.

In this setting, we found the row-normalization of the adjacency matrix to perform better:

$$\mathbf{P} = \frac{\tilde{\mathbf{A}} + \tilde{\mathbf{A}}^\top}{2n}, \text{ where } \tilde{A}_{ij} = A_{ij} \Big/ \sum_{k=1}^{n} A_{ik}. \tag{7}$$

Normalizing the adjacency matrix as in Equation (6) resulted in lower neighbor recalls and *k*NN accuracies, and in hedgehog-shaped embeddings with low-degree nodes pushed out to the periphery and dominating the embedding (Figure S4c,f). Furthermore, we experimented with various initialization schemes and found that on our graphs, random initialization performed very similar to Diffusion Maps (Figure S4b,e). This setting of graph NE can also be called graph *t*-SNE:

```
1 from openTSNE import TSNE
2 from openTSNE.affinity import PrecomputedAffinities
3 P = A / A.sum(axis=1)
4 P = (P + P.T) / 2 / A.shape[0]
5 Y = TSNE(initialization="spectral").fit(affinities=PrecomputedAffinities(P))
```

In contrast to the simple Equation (7) that we use for 2D graph NE, the closely related SGtSNEpi method (Pitsianis et al., 2019) derives the affinity matrix $\mathbf{P}$ from the adjacency matrix $\mathbf{A}$ in a more complicated way (Pitsianis et al., 2024, Supplementary). Non-zero elements $A_{ij}$ are first weighted by the Jaccard similarity of the sets of neighbors of nodes $i$ and $j$, then power-transformed to match a pre-specified row sum $\lambda$, and finally divided by $\lambda$ to yield $\tilde{\mathbf{A}}$. By default, $\lambda = 10$.

### 4.4 Graph NE with CNE backend approximates *t*-SNE backend

Node embeddings computed via `cne` and via `openTSNE` backends have the same optima:

**Theorem 4.1.** *[adapted from Ma & Collins 2018] Let $p$ be a probability distribution over $S = \{ij \mid 1 \leq i \neq j \leq n\}$, so that for all pairs $ij$, there is a path $p_{ik_1}, \ldots, p_{k_l j}$ with each step having positive probability. Let $w(\theta)$ be a family of non-negative functions $S \to \mathbb{R}_{\geq 0}$ parametrized by $\theta \in \Theta$ and symmetric in $i$ and $j$, meaning $w_{ij}(\theta) = w_{ji}(\theta)$. Let further $\xi$ be a probability distribution over $[n] := \{1, \ldots, n\}$ with full support. Suppose there is some $\theta^* \in \Theta$ and some $c > 0$ with $w(\theta^*)/c = p$. Then, $\theta^*$ minimizes the loss*

$$\mathcal{L}^{\text{InfoNCE}}(\theta) = -\mathbb{E}_{ij \sim p} \mathbb{E}_{k_1, \ldots, k_m \sim \xi} \log \frac{w_{ij}(\theta)/\xi_j}{w_{ij}(\theta)/\xi_j + \sum_{\alpha=1}^{m} w_{ik_\alpha}(\theta)/\xi_{k_\alpha}} \tag{8}$$

*and for any other minimizer $\tilde{\theta} \in \Theta$ there exists $\tilde{c} > 0$ with $w(\tilde{\theta})/\tilde{c} = p$.*

*In particular, the minima of $\mathcal{L}^{\text{InfoNCE}}$ correspond one-to-one to those of*

$$D_{\text{KL}}\big(p, w(\theta)/Z(\theta)\big) = -\mathbb{E}_{ij \sim p} \log \frac{w_{ij}(\theta)}{Z(\theta)} \tag{9}$$

*where $Z(\theta) = \sum_{ij} w_{ij}(\theta)$.*

The proof can be found in Appendix A. Consider this theorem in the case where $p$ is the uniform distribution of edges on a connected graph, parameters $\theta \in \mathbb{R}^{n \times d}$ are simply the embedding coordinates, and $w_{ij}(\theta)$ are the non-normalized low-dimensional affinities. Applying this theorem then shows that the InfoNCE loss

(when using a uniform noise distribution $\xi$, Equation 8) and the Kullback–Leibler divergence have the same minima.

This means that our graph NE framework unifies node embeddings and graph layouts. The main difference between graph NE with `cne` and `openTSNE` backends is the choice of optimization strategy for 128 and for 2 dimensions.

## 5 Experimental setup

**Datasets**  We used eight publicly available graph datasets (Table 1). The first five datasets were retrieved from the Deep Graph Library (Wang et al., 2019). The arXiv and MAG dataset were retrieved from the Open Graph Benchmark (Hu et al., 2020). The MNIST $k$NN dataset was obtained by computing the $k$NN graph with $k = 15$ on top of the 50 principal components of the MNIST digit dataset (LeCun et al., 1998). Each dataset was treated as an unweighted and undirected graph, where each node has a class label, used only for evaluation. We restricted ourselves to graphs with labeled nodes in order to use classification accuracy as one of the performance metrics. In all datasets we used only the largest connected component and excluded all self-loops if present, using `NetworkX` (Hagberg et al., 2008) functions `connected_components` and `selfloop_edges`. We did not use any node features.

| Dataset | Nodes | Edges | Classes | $E/N$ |
|---------|-------|-------|---------|-------|
| Citeseer | 2 120 | 7 358 | 6 | 3.5 |
| Cora | 2 485 | 10 138 | 7 | 4.1 |
| PubMed | 19 717 | 88 648 | 3 | 4.5 |
| Photo | 7 487 | 238 086 | 8 | 31.8 |
| Computer | 13 381 | 491 556 | 10 | 36.7 |
| MNIST $k$NN | 70 000 | 1 501 392 | 10 | 21.4 |
| arXiv | 169 343 | 2 315 598 | 40 | 13.7 |
| MAG | 726 664 | 10 778 888 | 349 | 14.8 |

Table 1: Benchmark datasets. Columns: number of nodes in the largest connected component, number of undirected edges, number of node classes, and the average number of edges per node.

**Performance metrics**  We evaluated the performance using three main metrics: neighbor recall, $k$NN classification accuracy, and linear classification accuracy. Such metrics are standard for evaluating graph embedding quality (Perozzi et al., 2014; Grover & Leskovec, 2016; Zhu et al., 2020a; Zhong et al., 2023). In addition to that, we used three further metrics: a metric based on the link-prediction task (Appendix B), top-$k$ 2-hop neighbor recall (Appendix C), and a Spearman correlation between shortest-path distances and embedding distances (on 1 000 random node pairs).

The neighbor recall quantifies how well local node neighborhoods are preserved in the embedding. We defined it as the average fraction of each node's graph neighbors that are among the node's nearest neighbors in the embedding:

$$\text{Recall} = \frac{1}{|\mathcal{V}|} \sum_{i=1}^{|\mathcal{V}|} \frac{\left| N_G[i] \cap N_{E,k_i}[i] \right|}{k_i}, \tag{10}$$

where $|\mathcal{V}|$ is the number of nodes in the graph, $N_G[i]$ is the set of node $i$'s graph neighbors, $k_i = |N_G[i]|$ is the number of node $i$'s graph neighbors, and $N_{E,k_i}[i]$ denotes the set of node $i$'s $k_i$ nearest neighbors in the embedding space. This metric does not require ground-truth classes and is similar to what is commonly used in the literature to benchmark graph-layout algorithms (Kruiger et al., 2017; Zhu et al., 2020a; Zhong et al., 2023). Therefore, we use this as our primary metric for measuring the embedding quality.

The $k$NN classification accuracy quantifies local class separation in the embedding. To calculate $k$NN accuracy, we randomly split all nodes into a training (90% of all nodes) and a test set (10%), and used the `KNeighborsClassifier` from scikit-learn (Pedregosa et al., 2011) with $k = 15$.

We used the cosine similarity for all $k$NN evaluations (recall and accuracy) in $d = 128$. CNE uses the cosine metric in its loss function (Equation 5), so only cosine neighbors make sense for evaluation. DeepWalk and

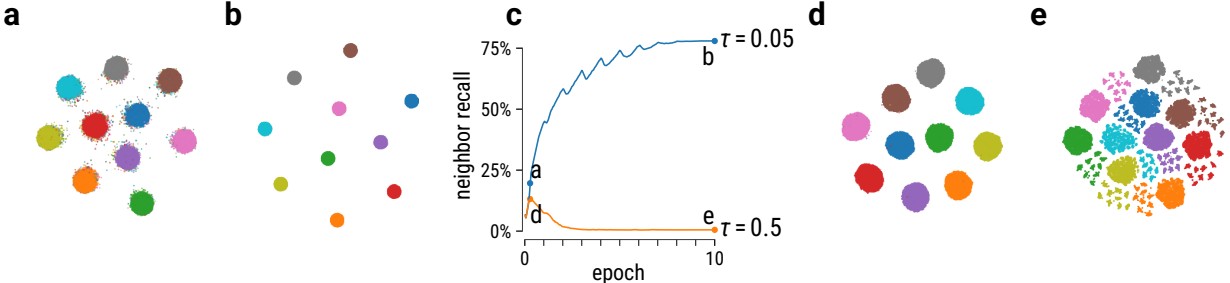

Figure 2: Learning dynamics of the 128-dimensional CNE embeddings of nodes in a stochastic-block-model graph with 10 blocks. **(a, b)** *t*-SNE visualizations of the 128D CNE embeddings with $\tau = 0.05$, during the first epoch and after ten epochs. **(c)** The neighbor recall as a function of the training epoch, for $\tau = 0.05$ and for $\tau = 0.5$. Labeled points correspond to *t*-SNE visualizations left/right. **(d, e)** Same as (a, b), but for $\tau = 0.5$.

node2vec rely on word2vec, which uses dot-product similarity in the loss function, and the original paper also used cosine metric for evaluation (Mikolov et al., 2013). In our experiments cosine evaluation led to better results on average for DeepWalk and node2vec. For all *k*NN evaluations in $d = 2$, we used the Euclidean distance.

For linear accuracy we used `LogisticRegression` from scikit-learn with no regularization (`penalty=None`), SAGA solver (Defazio et al., 2014) with `tol=0.01`, and the same train/test split. We standardized all features to have unit variance, based on the training set (as this speeds up convergence of the solver).

## 6 Node embeddings with graph NE require low temperature

In pilot experiments, we noticed that the node-embedding performance of graph NE (in 128D, with CNE backend) was strongly affected by the temperature parameter $\tau$. To investigate it further, we synthesized a graph following the stochastic block model (SBM; Holland et al., 1983). The generated graph had $80\,000$ nodes in 10 clusters, with any two nodes from the same cluster having probability $2.5 \cdot 10^{-3}$ to be connected by an edge, and any two nodes from two different clusters having probability $5 \cdot 10^{-6}$ to be connected. The resulting graph has a clear community structure that should be easy to recover.

CNE with the default temperature $\tau = 0.5$ achieved near-perfect class separation but failed to retain the neighborhood structure. The neighbor recall, after reaching 13% within the first training epoch, collapsed to below 1% over the next several epochs (Figure 2c, orange line). The *t*-SNE visualization of the high-dimensional embedding at the point of maximum neighbor recall showed ten compact clusters (Figure 2d), but after convergence it showed nine subclusters for each of the ten classes (Figure 2e). These smaller subclusters corresponded to nodes with an inter-cluster edge to a specific other class. During the optimization, these nodes got 'pulled out' of their class, destroying the local structure of the embedding and leading to near-zero neighbor recall.

In contrast, CNE with a lower temperature $\tau = 0.05$ did not show this behavior. The neighbor recall was almost monotonically increasing during training, reaching 78% after 10 epochs (Figure 2c, blue line). The *t*-SNE visualization showed ten compact clusters (Figure 2b), without any visible subclusters. Our interpretation is that the InfoNCE loss with low temperature could effectively ignore the noise in form of rare inter-class edges.

In the following experiments, we set the temperature of graph NE with CNE backend to $\tau = 0.05$ for all datasets. We have also implemented learnable temperature, making $\tau$ an additional trainable parameter. We found that on all our benchmark datasets, the temperature converged towards a value in a range of $[0.04, 0.08]$ (Table S7). In this setting, the evaluation results were close to the results with fixed $\tau = 0.05$ and both of them were usually much better than with $\tau = 0.5$. We report the performance for learned $\tau$, $\tau = 0.5$, and $\tau = 0.05$ in Tables S1 to S6.

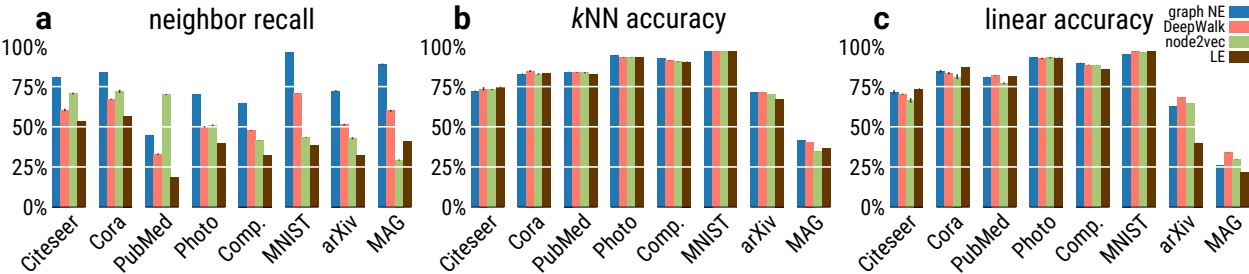

Figure 3: Performance metrics for node embeddings: **(a)** neighbor recall, **(b)** $k$NN accuracy, **(c)** linear accuracy. Datasets are ordered by the number of edges. For node2vec we did a grid search over $p, q \in \{0.25, 0.5, 1, 2, 4\}$ (Figure S2) and show results with the highest neighbor recall.

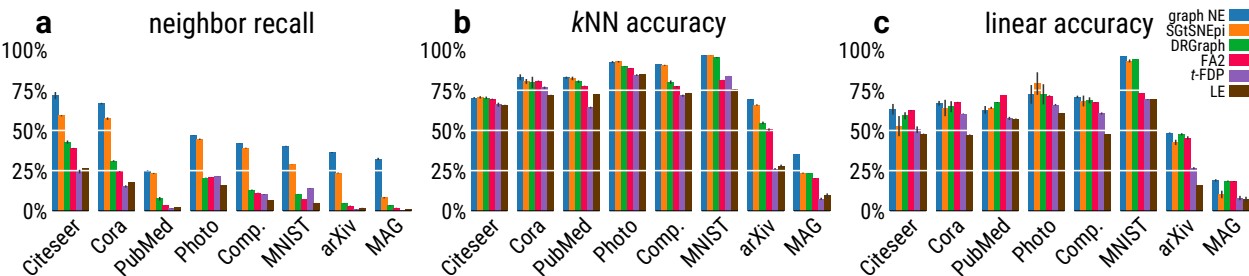

Figure 4: Performance metrics for 2D graph layouts: **(a)** neighbor recall, **(b)** $k$NN accuracy, **(c)** linear accuracy. See Figures 5 and S3 for the corresponding layouts.

Table 2: Neighbor recall for all methods and datasets (in %). All values are mean $\pm$ standard deviation across three training runs. The top performing method for each dimensionality is highlighted in bold. Methods in **blue** are ours. See Table S1 for additional graph NE variants in 128D.

| $d$ | Method | Citeseer | Cora | PubMed | Photo | Computer | MNIST | arXiv | MAG |
|---|---|---|---|---|---|---|---|---|---|
| 128 | **graph NE** | **81.0** $\pm$ 0.1 | **83.8** $\pm$ 0.0 | 44.3 $\pm$ 0.2 | **70.3** $\pm$ 0.1 | **64.8** $\pm$ 0.0 | **96.0** $\pm$ 0.0 | **72.3** $\pm$ 0.6 | **89.0** $\pm$ 0.5 |
| | DeepWalk | 60.5 $\pm$ 0.9 | 67.1 $\pm$ 0.4 | 32.9 $\pm$ 0.6 | 50.0 $\pm$ 0.5 | 47.7 $\pm$ 0.3 | 70.8 $\pm$ 0.2 | 51.4 $\pm$ 0.6 | 60.0 $\pm$ 0.7 |
| | node2vec | 70.7 $\pm$ 0.6 | 72.1 $\pm$ 1.0 | **70.1** $\pm$ 0.3 | 50.9 $\pm$ 0.5 | 41.3 $\pm$ 0.2 | 43.2 $\pm$ 0.2 | 42.9 $\pm$ 0.6 | 29.3 $\pm$ 0.4 |
| | Laplacian E. | 53.4 $\pm$ 0.0 | 56.7 $\pm$ 0.0 | 18.3 $\pm$ 0.1 | 39.7 $\pm$ 0.0 | 32.4 $\pm$ 0.0 | 38.5 $\pm$ 0.0 | 32.1 $\pm$ 0.1 | 40.6 $\pm$ 0.3 |
| 2 | **graph NE** | **71.7** $\pm$ 2.2 | **66.7** $\pm$ 0.5 | **25.0** $\pm$ 0.2 | **46.9** $\pm$ 0.2 | **41.8** $\pm$ 0.1 | **40.2** $\pm$ 0.2 | **36.3** $\pm$ 0.3 | **32.3** $\pm$ 0.7 |
| | SGtSNEpi | 59.1 $\pm$ 0.3 | 57.4 $\pm$ 0.8 | 23.3 $\pm$ 0.3 | 44.5 $\pm$ 0.4 | 39.0 $\pm$ 0.3 | 29.1 $\pm$ 0.1 | 23.6 $\pm$ 0.3 | 8.1 $\pm$ 0.4 |
| | DRGraph | 42.8 $\pm$ 1.0 | 31.0 $\pm$ 0.5 | 7.5 $\pm$ 1.1 | 20.5 $\pm$ 0.1 | 12.8 $\pm$ 0.4 | 10.0 $\pm$ 0.1 | 4.7 $\pm$ 0.2 | 3.2 $\pm$ 0.3 |
| | ForceAtlas2 | 38.8 $\pm$ 0.3 | 24.4 $\pm$ 0.3 | 3.2 $\pm$ 0.1 | 20.6 $\pm$ 0.1 | 11.1 $\pm$ 0.1 | 7.0 $\pm$ 0.0 | 2.9 $\pm$ 0.3 | 1.7 $\pm$ 0.2 |
| | $t$-FDP | 24.4 $\pm$ 1.2 | 15.2 $\pm$ 0.7 | 1.3 $\pm$ 0.1 | 21.6 $\pm$ 0.1 | 10.2 $\pm$ 0.2 | 13.9 $\pm$ 0.1 | 0.7 $\pm$ 0.2 | 0.3 $\pm$ 0.1 |
| | Laplacian E. | 26.2 $\pm$ 0.1 | 17.9 $\pm$ 0.0 | 2.0 $\pm$ 0.1 | 16.0 $\pm$ 0.0 | 6.4 $\pm$ 0.0 | 5.0 $\pm$ 0.0 | 1.6 $\pm$ 0.3 | 0.8 $\pm$ 0.1 |

## 7 Benchmarking graph NE

### 7.1 Graph NE outperforms other node embeddings

We compared graph NE with the popular non-parametric node-embedding algorithms DeepWalk (Perozzi et al., 2014) and node2vec (Grover & Leskovec, 2016), as well as with Laplacian Eigenmaps (LE), all optimizing 128-dimensional embeddings. We used node2vec's implementation from PyTorch Geometric (Fey & Lenssen, 2019) and the DeepWalk implementation from DGL (Wang et al., 2019). We ran both methods with the default parameters for 100 epochs (as we did for CNE, see Figure S1 for runtimes). For node2vec, we ran a sweep over the parameters $p, q \in \{0.25, 0.5, 1, 2, 4\}$, as in the original paper, and report the results with the highest neighbor recall (for all results, see Figure S2). For LE we used scikit-learn (Pedregosa et al., 2011), with LOBPCG (Knyazev et al., 2007) for solving the generalized eigenproblem.

We found that graph NE outperformed the other algorithms in terms of neighbor recall on seven datasets out of eight; the only exception was the PubMed dataset (Figure 3a, Table 2). Across all datasets, the average gap in neighbor recall between graph NE and the best other method was 13.4 percentage points, showing a strong improvement over competitors. In terms of the top-k 2-hop neighbor recall (Appendix C), our graph NE outperformed the competitors on all datasets apart from the Photo graph (Table S6).

In terms of the classification accuracies, the results on most datasets were very similar across all methods. Graph NE had slightly lower $k$NN accuracy on the two smallest datasets (Citeseer and Cora), and was the best or within 1% of the best on all other datasets (Figure 3b, Table S2). In terms of linear accuracy, graph NE yielded competitive results and lagged only slightly behind other methods for some datasets (Figure 3c, Table S3). Curiously, graph NE with $\tau = 0.5$ was the best or within 1% of the best on all datasets apart from Cora and MAG, where it was slightly behind (Table S3); but this temperature led to substantially worse neighbor recall (Table S1). This suggests a trade-off between linear classification and neighbor quality.

Graph NE outperformed the other methods on the link prediction task (Table S4) but the performance for many methods was close to saturated 100%. For that reason we prefer the $k$NN recall metric which is conceptually similar (Appendix B). Graph NE also outperformed all other methods in terms of Spearman correlation between the shortest-path distances and the embedding distances, on all datasets apart from MAG, where Laplacian Eigenmaps showed the best results (Figure S5).

In summary, results in terms of classification accuracies were all similar, but neighbor recall showed large and pronounced differences with graph NE performing the best by a large margin.

## 7.2 Graph NE outperforms other graph layouts in terms of local structure

We benchmarked graph NE with the `openTSNE` backend against five existing graph-layout algorithms: SGtSNEpi Pitsianis et al. (2019), ForceAtlas2 (FA2; Jacomy et al., 2014), Laplacian Eigenmaps (LE; Belkin & Niyogi, 2003), DRGraph (Zhu et al., 2020a), and $t$-FDP (Zhong et al., 2023). We also performed comparisons with Diffusion Maps (Coifman & Lafon, 2006, with $t = 1$ diffusion step), which differ from Laplacian Eigenmaps only by scaling, but found that they produced results very similar to Laplacian Eigenmaps, so we do not report them separately. We did not include tsNET (Kruiger et al., 2017), because it cannot embed large graphs.[1] Unless specified otherwise, we used the original implementation of the algorithms and ran them with the default parameters. For FA2 we used the Barnes–Hut implementation by Chippada (2017). Both $t$-SNE and $t$-FDP are implemented in Cython, DRGraph and SGtSNEpi are implemented in C++ and offer wrappers in Python and Julia, respectively. For consistency, we used Diffusion Maps initialization for all algorithms where possible (all except SGtSNEpi and DRGraph).

Graph NE showed outstanding performance on all of our benchmark datasets. The neighbor recall of graph NE was always the highest, with SGtSNEpi only sometimes coming close (Figure 4a, Table 2). In the top-k 2-hop neighbor recall that focuses on pairs of nodes that share many neighbors (Appendix C), graph NE also performed on average the best (Table S6). In this metric, SGtSNEpi only marginally (within 1%) outperformed graph NE on two graphs and was much worse on several other graphs.

In terms of $k$NN accuracy, graph NE was either the top performing method or within 1% of the top performing method for all datasets (Figure 4b, Table S2). In terms of linear accuracy the same was true for six out of the eight datasets (Figure 4c, Table S3).

In terms of the Spearman correlation between the shortest-path distances and the embedding distances, consistent winners across datasets were ForceAtlas2 and $t$-FDP (Table S5). This is likely related to the fact that in 2D embeddings, there is a trade-off between preserving local and global structure (Böhm et al., 2022).

Qualitatively, graph NE layouts performed well in terms of separating clusters from each other and bringing out sub-cluster details within individual clusters (Figure 5).

---

[1] While our paper was in print, Meidiana et al. (2025) developed a fast implementation of tsNET. We leave its benchmarking for future work. In parallel, Meidiana & Hong (2025) studied UMAP applied to the pairwise shortest-path distances between nodes, like in tsNET.

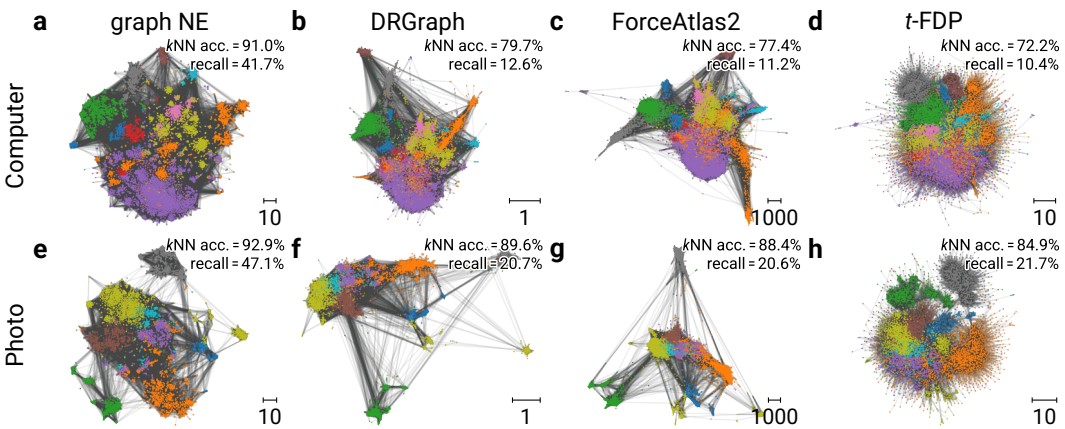

Figure 5: Embeddings of the Computer and Photo datasets obtained using our graph NE (graph *t*-SNE), DRGraph, ForceAtlas2, and *t*-FDP. Embeddings were aligned using Procrustes rotation. See Figure S3 for all datasets and methods.

## 8    Discussion

We suggested *graph NE*, a novel approach to non-parametric node embeddings, and showed that it outperforms existing competitors in terms of preserving local graph structure, both for high-dimensional embeddings and for 2D graph layouts. Graph NE can be efficiently implemented using existing neighbor-embedding backends, CNE and openTSNE. Both backends scale linearly in the number of graph edges and achieve competitive runtimes for large graphs. For small graphs, we observed that the openTSNE backend was slower than some of the competitors (Figure S1).

In this work, we focused on complex real-world graphs and have purposefully not tested our graph NE on simple planar graphs or 3D mesh graphs that are often used for benchmarking graph layout algorithms. Such graphs are arguably not an interesting case for high-dimensional embeddings, and we aimed to use the same graphs for all of our benchmarks.

Our work opens up several directions for future work. First, CNE allows to train parametric embeddings (Damrich et al., 2023), which we have not explored here. How would parametric CNE with a GCN mapping compare to existing GCL methods, in particular augmentation-free methods? Parametric models need to use node features. Given the ongoing debate about the usefulness of combining node and edge information in graph learning (Errica et al., 2020; Faber et al., 2021; Bechler-Speicher et al., 2024; Coupette et al., 2025), it would be interesting to study how much node features can help with node embeddings.

Second, we only used *t*-SNE-like losses here, but a similar approach could be implemented using other neighbor-embedding algorithms, e.g. UMAP (McInnes et al., 2018). How would graph UMAP (2D and high-D) perform for graph layouts and node embeddings, especially in contrast to DRgraph and DeepWalk/node2vec, which, like UMAP, use negative sampling for optimization?

Third, our results point to a non-trivial effect that the temperature parameter $\tau$ can have on InfoNCE-based embeddings (Figure 2). Further investigation of this phenomenon and its potential relevance for contrastive learning in computer vision and other domains also remains for future work.

Our graph NE algorithms succeed *because* of their simplicity, not *despite* of it. The straightforward loss function enables efficient optimization strategies, which scale linearly with the graph size and preserve nodes' neighbors better than other algorithms.

## Code availability

Our code is available at `https://github.com/berenslab/graph-ne-paper`.

## Acknowledgements

The authors would like to thank Leland McInnes and Pavlin Poličar for discussions, and Philipp Berens for support and feedback. The authors benefited from the discussions at the Dagstuhl seminar 24122 supported by the Leibniz Center for Informatics.

This work was partially funded by the Gemeinnützige Hertie-Stiftung, the Cyber Valley Research Fund (D.30.28739), the Danmarks Frie Forskningsfond (Sapere Aude 1051-00106B), and the National Institutes of Health (UM1MH130981). The content is solely the responsibility of the authors and does not necessarily represent the official views of the National Institutes of Health. Dmitry Kobak is a member of the Germany's Excellence cluster 2064 "Machine Learning — New Perspectives for Science" (EXC 390727645). The authors thank the International Max Planck Research School for Intelligent Systems (IMPRS-IS) for supporting Jan Niklas Böhm.

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

# Appendix

## A  Proof of Theorem 1

**Theorem 4.1.** *[adapted from Ma & Collins 2018] Let $p$ be a probability distribution over $S = \{ij \mid 1 \leq i \neq j \leq n\}$, so that for all pairs $ij$, there is a path $p_{ik_1}, \ldots, p_{k_l j}$ with each step having positive probability. Let $w(\theta)$ be a family of non-negative functions $S \to \mathbb{R}_{\geq 0}$ parametrized by $\theta \in \Theta$ and symmetric in $i$ and $j$, meaning $w_{ij}(\theta) = w_{ji}(\theta)$. Let further $\xi$ be a probability distribution over $[n] := \{1, \ldots, n\}$ with full support. Suppose there is some $\theta^* \in \Theta$ and some $c > 0$ with $w(\theta^*)/c = p$. Then, $\theta^*$ minimizes the loss*

$$\mathcal{L}^{\mathrm{InfoNCE}}(\theta) = -\mathbb{E}_{ij \sim p}\mathbb{E}_{k_1, \ldots, k_m \sim \xi} \log \frac{w_{ij}(\theta)/\xi_j}{w_{ij}(\theta)/\xi_j + \sum_{\alpha=1}^{m} w_{ik_\alpha}(\theta)/\xi_{k_\alpha}} \tag{8}$$

*and for any other minimizer $\tilde{\theta} \in \Theta$ there exists $\tilde{c} > 0$ with $w(\tilde{\theta})/\tilde{c} = p$.*

*In particular, the minima of $\mathcal{L}^{\mathrm{InfoNCE}}$ correspond one-to-one to those of*

$$D_{\mathrm{KL}}\big(p, w(\theta)/Z(\theta)\big) = -\mathbb{E}_{ij \sim p} \log \frac{w_{ij}(\theta)}{Z(\theta)} \tag{9}$$

*where $Z(\theta) = \sum_{ij} w_{ij}(\theta)$.*

*Proof.* The key idea is to rewrite the loss as an average over cross-entropy losses over the set $\{0, \ldots, m\}$. For $i, k_0, \ldots, k_m \in \{1, \ldots, n\}$ define

$$\alpha(i, k_0, \ldots, k_m) := \sum_{\mu=0}^{m} \left( p_{ik_\mu} \prod_{\nu=0, \nu\neq\mu}^{m} \xi_{k_\nu} \right) \tag{11}$$

$$\beta(\mu \mid i, k_0, \ldots, k_m) := \frac{p_{ik_\mu} \prod_{\nu=0,\nu\neq\mu}^{m} \xi_{k_\nu}}{\alpha(i, k_0, \ldots, k_m)} = \frac{p_{ik_\mu}/\xi_{k_\mu}}{\sum_{\nu=0}^{m} p_{ik_\nu}/\xi_{k_\nu}} \tag{12}$$

$$\gamma_\theta(\mu \mid i, k_0, \ldots, k_m) := \frac{w_{ik_\mu}(\theta)/\xi_{k_\mu}}{\sum_{\nu=0}^{m} w_{ik_\nu}(\theta)/\xi_{k_\nu}} \tag{13}$$

Note that $\alpha(i, k_0, \ldots, k_m)$ can only be zero if $p_{ik_\mu} = 0$ for all $\mu = 0, \ldots, m$. In this case, we define $\beta(\mu \mid i, k_0, \ldots, k_m) = 0$ for all $\mu = 0, \ldots, m$. If all $w_{ik_\nu}(\theta)$ are zero for $\nu = 0, \ldots, m$, we define $\gamma_\theta(\mu \mid i, k_0, \ldots, k_m) = 0$. If they are not identical to zero, $\beta(\cdot \mid i, k_0, \ldots, k_m)$ and $\gamma_\theta(\cdot \mid i, k_0, \ldots, k_m)$ are probability distributions over $\{0, \ldots, m\}$.

By the proof of Theorem 4.1 in Ma & Collins (2018), we can rewrite

$$\mathcal{L}^{\mathrm{InfoNCE}}(\theta) = \sum_{i, k_0, \ldots, k_m \in [n]} \frac{\alpha(i, k_0, \ldots, k_m)}{m+1} \left( -\sum_{\mu=0}^{m} \beta(\mu \mid i, k_0, \ldots, k_m) \log \gamma_\theta(\mu \mid i, k_0, \ldots, k_m) \right) \tag{14}$$

The latter term in parentheses is a cross-entropy loss over probability distributions over the set $\{0, \ldots, m\}$. It becomes minimal if and only if $\beta(\cdot \mid i, k_0, \ldots, k_m) = \gamma_\theta(\cdot \mid i, k_0, \ldots, k_m)$. Since $\alpha(i, k_0, \ldots, k_m)$ is positive as soon as $\min_\mu p_{ik_\mu} > 0$ for some $\mu$, we conclude that the minima of $\mathcal{L}^{\mathrm{InfoNCE}}(\theta)$ are those $\theta$ for which $\beta(\cdot \mid i, k_0, \ldots, k_m) = \gamma_\theta(\cdot \mid i, k_0, \ldots, k_m)$ whenever $\min_\mu p_{ik_\mu} > 0$.

Clearly $\theta^*$ is a minimum of $\mathcal{L}^{\mathrm{InfoNCE}}(\theta)$ as

$$\beta(\cdot \mid i, k_0, \ldots, k_m) = \gamma_{\theta^*}(\cdot \mid i, k_0, \ldots, k_m)$$

holds for all $i, k_0, \ldots, k_m \in [n]$.

Conversely, let $\tilde{\theta}$ be a minimizer of $\mathcal{L}^{\text{InfoNCE}}(\theta)$. Consider $ij$ with $p_{ij} > 0$ and arbitrary $k \in [n]$. Taking $k_0 = j$ and $k_1, \ldots, k_m = k$, we have

$$\beta(0 \mid i, k_0 = j, k_1 = k, \ldots, k_m = k) = \gamma_{\tilde{\theta}}(0 \mid i, k_0 = j, k_1 = k, \ldots, k_m = k) \tag{15}$$

$$\Leftrightarrow \frac{p_{ij}/\xi_j}{p_{ij}/\xi_j + m p_{ik}/\xi_k} = \frac{w_{ij}(\tilde{\theta})/\xi_j}{w_{ij}(\theta)/\xi_j + m w_{ik}(\tilde{\theta})/\xi_k} \quad \text{(hence } w_{ij}(\tilde{\theta}) > 0) \tag{16}$$

$$\Leftrightarrow \frac{1}{1 + m \frac{p_{ik}\xi_j}{p_{ij}\xi_k}} = \frac{1}{1 + m \frac{w_{ik}(\tilde{\theta})\xi_j}{w_{ij}(\tilde{\theta})\xi_k}} \tag{17}$$

$$\Leftrightarrow \frac{p_{ik}}{p_{ij}} = \frac{w_{ik}(\tilde{\theta})}{w_{ij}(\tilde{\theta})} \tag{18}$$

$$\Rightarrow \frac{\sum_k p_{ik}}{p_{ij}} = \frac{\sum_k w_{ik}(\tilde{\theta})}{w_{ij}(\tilde{\theta})} \tag{19}$$

$$\Rightarrow \frac{w_{ij}(\tilde{\theta})}{p_{ij}} = \frac{\sum_k w_{ik}(\tilde{\theta})}{\sum_k p_{ik}} =: c_i, \tag{20}$$

which only depends on $i$, not on $j$. By symmetry of $w$, we have for any $ij$ with $p_{ij} > 0$ that $p_{ji} > 0$, too, and

$$c_i = \frac{w_{ij}(\tilde{\theta})}{p_{ij}} = \frac{w_{ij}(\tilde{\theta})}{w_{ij}(\theta^*)} = \frac{w_{ji}(\tilde{\theta})}{w_{ji}(\theta^*)} = \frac{w_{ji}(\tilde{\theta})}{p_{ji}} = c_j.$$

Now we use the assumption that one can get from any $i$ to any $j$ via transitions with positive $p$. This implies that for each $i$, there is some $j$ with $p_{ij} > 0$ (which we ensure in our experiments by only considering a connected component, Section 5). Moreover, for any $i, j$ we get $c_i = c_j =: c$. So, we have for any $ij$ with $p_{ij} > 0$ that $p_{ij} = w_{ij}(\tilde{\theta})/c$.

Finally, let $ij$ be such that $p_{ij} = 0$. By the path connectedness assumption, there is some $k \in [n]$ with $p_{ik} > 0$. Taking $k_0 = k$ and $k_1, \ldots, k_m = j$ we get as above that

$$1 = \frac{p_{ik}}{p_{ik} + m p_{ij}} = \frac{w_{ik}(\tilde{\theta})}{w_{ik}(\tilde{\theta}) + m w_{ij}(\tilde{\theta})} \tag{21}$$

from which we can conclude that $w_{ij}(\tilde{\theta})$ must be zero, whenever $p_{ij}$ is zero. This implies $p = w(\tilde{\theta})/c$.

It is well-known that the KL divergence between two probability distributions is minimal if and only if they are equal. $\qquad\square$

Our setup in Theorem 4.1 deviates from that of Theorem 4.1 in Ma & Collins (2018), because they only show a statement along the lines of Equation (20). In contrast, we show that the InfoNCE loss has same minima as the KL divergence using stronger assumptions (Ma & Collins's Assumption 2.2, symmetry, and the path connectedness with positive probability).

## B   Link prediction as evaluation metric

Link prediction is one of the standard evaluation criteria in the graph-learning community (Kipf & Welling, 2017). In a link-prediction task, the predictive model is given pairs of nodes which may or may not be connected by an edge and must rank these pairs by the likelihood of predicting an edge correctly (Zhang & Chen, 2018). We used 10% of all graph edges as positive test pairs and equally many non-edges as negative test pairs. We scored each of the test pairs based on their embedding points, either by cosine similarity (128D) or by negative Euclidean distance (2D) and computed the area under the ROC curve, similar to the setup from Grover & Leskovec (2016).

The neighbor recall metric is conceptually similar to link prediction. It is the recall of a directed link predictor $P$ that predicts the links from node $i$ to the $k_i$ nearest nodes to $i$'s, where $k_i$ is the degree of node

$i$. In the case of constant, known node degree $k$, the recall $r$ is linearly related to the accuracy $a$ of $P$ by $a = 1 - (1 - r) \cdot 2k/(n - 1)$. As the task of link prediction is easier and saturates faster compared to the neighbor recall (Table S1 vs. Table S4), we use neighbor recall as our main metric.

## C  Top-k 2-hop neighbor recall as evaluation metric

Our neighbor recall metric, defined in Equation (10), treats all neighbors of a node equally. In some cases, it may be desirable to focus on node pairs that have many *shared* neighbors, and make sure that they are embedded close together. We quantified this using an additional metric, which we call *top-k 2-hop neighbor recall*.

We first compute number of shared neighbors for any pair of nodes $i, j$:

$$S(i, j) = \big| N_G[i] \cap N_G[j] \big|, \tag{22}$$

where $N_G[i]$ is the set of neighbors of node $i$ in graph $G$. Be definition, the $S(i, \cdot)$ is non-zero for all 2-hop neighbors of node $i$. Note that $S(i, j) / \big| N_G[i] \cup N_G[j] \big|$ gives Jaccard similarity between $i$ and $j$.

Now let $N_{S,k}[i]$ denote the set of $k$ nodes with the highest values of $S(i, \cdot)$. The top-k 2-hop neighbor recall is the fraction of these nodes contained among the $k$ closest embedding points $N_{E,k}[i]$ of node $i$:

$$\text{Top-k 2-hop recall} = \frac{1}{|\mathcal{V}|} \sum_{i=1}^{|\mathcal{V}|} \frac{\big| N_{S,k}[i] \cap N_{E,k}[i] \big|}{k}. \tag{23}$$

We used $k = 10$ and report the results in Table S6. For nodes that had fewer than $k$ other nodes with non-zero $S(i, \cdot)$, the set $N_{S,k}[i]$ contained fewer than $k$ elements.

# D   Supplementary Figures and Tables

Table S1: Neighbor recall for all methods and datasets (in %). All values are mean ± standard deviation across three training runs. The top performing method for each dimensionality (and all methods within 1%) is highlighted in bold. Methods in **blue** are ours. "Graph NE$\tau$" means that the temperature $\tau$ was learned as a parameter during training, see Section 6.

| $d$ | Method | Citeseer | Cora | PubMed | Photo | Computer | MNIST | arXiv | MAG |
|---|---|---|---|---|---|---|---|---|---|
| 128 | **graph NE** | **81.0** ± 0.1 | **83.8** ± 0.0 | 44.3 ± 0.2 | **70.3** ± 0.1 | **64.8** ± 0.0 | **96.0** ± 0.0 | **72.3** ± 0.5 | 89.0 ± 0.4 |
|  | **graph NE$\tau$** | 80.3 ± 0.0 | 82.8 ± 0.1 | 42.3 ± 0.1 | 69.5 ± 0.0 | 64.0 ± 0.1 | **96.0** ± 0.0 | **72.5** ± 0.7 | **91.0** ± 0.1 |
|  | CNE, $\tau = 0.5$ | 55.9 ± 0.1 | 58.1 ± 0.0 | 21.9 ± 0.3 | 35.7 ± 0.1 | 31.9 ± 0.0 | 39.2 ± 0.0 | 34.4 ± 0.2 | 33.2 ± 0.2 |
|  | DeepWalk | 60.5 ± 0.7 | 67.1 ± 0.3 | 32.9 ± 0.5 | 50.0 ± 0.4 | 47.7 ± 0.3 | 70.8 ± 0.2 | 51.4 ± 0.5 | 60.0 ± 0.6 |
|  | node2vec | 70.7 ± 0.5 | 72.1 ± 1.0 | **70.1** ± 0.3 | 50.9 ± 0.5 | 41.3 ± 0.2 | 43.2 ± 0.2 | 42.9 ± 0.6 | 29.3 ± 0.3 |
|  | Laplacian E. | 53.4 ± 0.0 | 56.7 ± 0.0 | 18.3 ± 0.1 | 39.7 ± 0.0 | 32.4 ± 0.0 | 38.5 ± 0.0 | 32.1 ± 0.1 | 40.6 ± 0.2 |
| 2 | **graph NE** | **71.7** ± 1.8 | **66.7** ± 0.4 | **25.0** ± 0.1 | **46.9** ± 0.1 | **41.8** ± 0.1 | **40.2** ± 0.2 | **36.3** ± 0.3 | **32.3** ± 0.6 |
|  | SGtSNEpi | 59.1 ± 0.3 | 57.4 ± 0.6 | 23.3 ± 0.2 | 44.5 ± 0.3 | 39.0 ± 0.2 | 29.1 ± 0.1 | 23.6 ± 0.2 | 8.1 ± 0.3 |
|  | DRGraph | 42.8 ± 0.8 | 31.0 ± 0.4 | 7.5 ± 0.9 | 20.5 ± 0.1 | 12.8 ± 0.3 | 10.0 ± 0.1 | 4.7 ± 0.1 | 3.2 ± 0.2 |
|  | ForceAtlas2 | 38.8 ± 0.3 | 24.4 ± 0.3 | 3.2 ± 0.1 | 20.6 ± 0.1 | 11.1 ± 0.1 | 7.0 ± 0.0 | 2.9 ± 0.3 | 1.7 ± 0.1 |
|  | $t$-FDP | 24.4 ± 0.9 | 15.2 ± 0.6 | 1.3 ± 0.1 | 21.6 ± 0.1 | 10.2 ± 0.2 | 13.9 ± 0.1 | 0.7 ± 0.1 | 0.3 ± 0.1 |
|  | Laplacian E. | 26.2 ± 0.1 | 17.9 ± 0.0 | 2.0 ± 0.1 | 16.0 ± 0.0 | 6.4 ± 0.0 | 5.0 ± 0.0 | 1.6 ± 0.2 | 0.8 ± 0.1 |

Table S2: $k$NN classification accuracy. The same setup as in Table S1, with random training/test splits used for each run.

| $d$ | Method | Citeseer | Cora | PubMed | Photo | Computer | MNIST | arXiv | MAG |
|---|---|---|---|---|---|---|---|---|---|
| 128 | **graph NE** | 72.0 ± 0.4 | 82.7 ± 0.0 | **84.1** ± 0.1 | **94.3** ± 0.1 | **92.4** ± 0.1 | **97.2** ± 0.0 | **71.3** ± 0.1 | **41.6** ± 0.1 |
|  | **graph NE$\tau$** | 72.2 ± 0.4 | 83.1 ± 0.3 | **83.8** ± 0.3 | **94.3** ± 0.1 | **92.4** ± 0.2 | **97.1** ± 0.0 | **71.7** ± 0.1 | **41.7** ± 0.1 |
|  | CNE, $\tau = 0.5$ | 72.8 ± 0.2 | 83.3 ± 0.2 | **83.1** ± 0.0 | 92.6 ± 0.0 | 91.4 ± 0.0 | 96.9 ± 0.0 | **71.1** ± 0.1 | 36.5 ± 0.1 |
|  | DeepWalk | **73.6** ± 1.0 | **84.7** ± 0.6 | **83.7** ± 0.2 | 93.3 ± 0.1 | 91.1 ± 0.2 | **97.0** ± 0.1 | **71.2** ± 0.1 | 40.5 ± 0.0 |
|  | node2vec | 73.1 ± 0.4 | 82.8 ± 0.5 | **83.6** ± 0.4 | 93.1 ± 0.2 | 90.5 ± 0.2 | **96.8** ± 0.0 | 70.1 ± 0.0 | 34.4 ± 0.1 |
|  | Laplacian E. | **74.5** ± 0.0 | 83.1 ± 0.0 | 82.6 ± 0.0 | 93.0 ± 0.0 | 90.3 ± 0.0 | **96.7** ± 0.0 | 67.2 ± 0.1 | 36.5 ± 0.0 |
| 2 | **graph NE** | **70.3** ± 0.4 | **83.1** ± 1.5 | **82.9** ± 0.5 | **92.6** ± 0.5 | **91.0** ± 0.0 | **96.8** ± 0.1 | **69.4** ± 0.2 | **35.3** ± 0.0 |
|  | SGtSNEpi | **70.6** ± 0.6 | 80.5 ± 1.3 | **82.4** ± 0.9 | **92.8** ± 0.4 | **90.6** ± 0.3 | **96.9** ± 0.1 | 65.9 ± 0.3 | 23.5 ± 0.3 |
|  | DRGraph | **70.3** ± 0.7 | 79.8 ± 2.9 | 80.4 ± 0.6 | 89.8 ± 0.2 | 80.1 ± 0.9 | 95.2 ± 0.3 | 54.7 ± 0.7 | 23.3 ± 0.1 |
|  | ForceAtlas2 | 69.3 ± 0.4 | 80.6 ± 0.3 | 77.6 ± 0.2 | 88.6 ± 0.2 | 77.2 ± 0.4 | 81.4 ± 0.0 | 50.6 ± 0.6 | 20.0 ± 0.2 |
|  | $t$-FDP | 66.0 ± 1.0 | 76.6 ± 0.6 | 64.2 ± 0.5 | 84.5 ± 0.4 | 71.6 ± 0.5 | 83.9 ± 0.0 | 26.1 ± 0.4 | 7.5 ± 0.6 |
|  | Laplacian E. | 65.6 ± 0.0 | 71.8 ± 0.0 | 72.4 ± 0.2 | 84.8 ± 0.2 | 73.2 ± 0.2 | 75.3 ± 0.1 | 27.8 ± 0.8 | 9.4 ± 1.2 |

Table S3: Linear classification accuracy. The same setup as in Table S2 applies.

| $d$ | Method | Citeseer | Cora | PubMed | Photo | Computer | MNIST | arXiv | MAG |
|---|---|---|---|---|---|---|---|---|---|
| 128 | **graph NE** | 71.5 ± 1.2 | 84.3 ± 1.0 | 80.9 ± 0.2 | **93.0** ± 0.2 | **89.7** ± 0.2 | 95.3 ± 0.0 | 62.6 ± 0.2 | 26.1 ± 0.1 |
|  | **graph NE$\tau$** | 71.9 ± 1.4 | 83.6 ± 0.7 | 80.2 ± 0.7 | **93.4** ± 0.4 | **89.5** ± 0.4 | 90.8 ± 0.3 | 63.6 ± 0.1 | 27.6 ± 0.1 |
|  | CNE, $\tau = 0.5$ | **72.8** ± 0.8 | 84.3 ± 0.7 | **83.6** ± 0.3 | 92.7 ± 0.2 | **89.4** ± 0.2 | **97.1** ± 0.0 | **67.9** ± 0.2 | 32.2 ± 0.0 |
|  | DeepWalk | 70.3 ± 0.0 | 83.3 ± 0.7 | 81.9 ± 0.3 | **92.5** ± 0.4 | 88.5 ± 0.2 | **96.7** ± 0.1 | **68.1** ± 0.0 | **34.2** ± 0.1 |
|  | node2vec | 66.4 ± 1.4 | 81.0 ± 1.4 | 77.0 ± 0.5 | **93.0** ± 0.5 | 88.5 ± 0.1 | **96.1** ± 0.1 | 64.6 ± 0.0 | 29.9 ± 0.2 |
|  | Laplacian E. | **73.6** ± 0.0 | **86.7** ± 0.0 | 81.4 ± 0.0 | **92.8** ± 0.0 | 85.5 ± 0.0 | **97.0** ± 0.0 | 40.0 ± 0.1 | 21.3 ± 0.1 |
| 2 | **graph NE** | **63.2** ± 2.7 | **66.9** ± 1.2 | 62.8 ± 2.0 | 72.5 ± 4.8 | **70.9** ± 0.7 | **96.0** ± 0.1 | **48.4** ± 0.3 | **18.9** ± 0.6 |
|  | SGtSNEpi | 52.7 ± 5.2 | 64.0 ± 4.2 | 64.1 ± 0.5 | **79.2** ± 5.8 | 68.3 ± 2.8 | 93.3 ± 0.7 | 42.6 ± 1.4 | 10.2 ± 1.9 |
|  | DRGraph | 59.1 ± 1.8 | 64.8 ± 2.8 | 67.4 ± 0.2 | 72.5 ± 5.2 | 68.8 ± 1.5 | 94.2 ± 0.1 | **47.6** ± 0.6 | **18.5** ± 0.3 |
|  | ForceAtlas2 | **62.4** ± 0.2 | **67.3** ± 0.0 | **71.6** ± 0.0 | 71.4 ± 0.4 | 67.3 ± 0.2 | 73.1 ± 0.0 | 45.4 ± 0.8 | **18.2** ± 0.3 |
|  | $t$-FDP | 50.5 ± 1.7 | 60.3 ± 0.2 | 57.5 ± 0.8 | 66.0 ± 0.4 | 60.6 ± 0.5 | 69.1 ± 0.2 | 26.5 ± 0.5 | 8.0 ± 1.1 |
|  | Laplacian E. | 47.6 ± 0.0 | 47.2 ± 0.0 | 57.1 ± 0.0 | 60.4 ± 0.0 | 47.3 ± 0.0 | 69.4 ± 0.0 | 15.6 ± 0.0 | 7.1 ± 1.4 |

Table S4: Area under the link prediction ROC curve. The same setup as in Table S2. See Appendix B.

| $d$ | Method | Citeseer | Cora | PubMed | Photo | Computer | MNIST | arXiv | MAG |
|---|---|---|---|---|---|---|---|---|---|
| 128 | **graph NE** | **100.0** ± 0.0 | **100.0** ± 0.0 | **100.0** ± 0.0 | 99.7 ± 0.0 | **99.4** ± 0.0 | **100.0** ± 0.0 | **100.0** ± 0.0 | **100.0** ± 0.0 |
| | **graph NE**$\tau$ | **100.0** ± 0.0 | **100.0** ± 0.0 | **100.0** ± 0.0 | **99.6** ± 0.0 | 99.3 ± 0.0 | **100.0** ± 0.0 | **100.0** ± 0.0 | **100.0** ± 0.0 |
| | CNE, $\tau = 0.5$ | 99.7 ± 0.0 | 99.7 ± 0.0 | 99.8 ± 0.0 | 97.5 ± 0.0 | 96.2 ± 0.0 | 99.8 ± 0.0 | 98.7 ± 0.0 | 99.7 ± 0.0 |
| | DeepWalk | 99.8 ± 0.0 | 99.8 ± 0.0 | 99.9 ± 0.0 | 98.1 ± 0.1 | 97.2 ± 0.1 | **100.0** ± 0.0 | 99.5 ± 0.0 | **100.0** ± 0.0 |
| | node2vec | 91.6 ± 0.3 | 88.3 ± 0.2 | 84.3 ± 0.3 | 83.3 ± 0.2 | 71.9 ± 0.2 | 99.9 ± 0.0 | 66.3 ± 0.2 | 93.7 ± 0.1 |
| | Laplacian E. | 99.8 ± 0.0 | 99.7 ± 0.0 | 98.9 ± 0.0 | 97.8 ± 0.0 | 96.0 ± 0.0 | 99.8 ± 0.0 | 95.3 ± 0.0 | 99.2 ± 0.0 |
| 2 | **graph NE** | 98.2 ± 0.1 | 96.6 ± 0.2 | 96.3 ± 0.3 | **96.4** ± 0.1 | 94.1 ± 0.0 | 98.4 ± 0.0 | 95.7 ± 0.2 | 98.4 ± 0.0 |
| | SGtSNEpi | 97.6 ± 0.2 | 95.5 ± 0.2 | 96.0 ± 0.3 | 96.1 ± 0.1 | 93.8 ± 0.2 | 98.2 ± 0.1 | 95.3 ± 0.1 | 96.4 ± 0.1 |
| | DRGraph | **98.9** ± 0.0 | 97.6 ± 0.1 | **97.6** ± 0.0 | **96.4** ± 0.2 | 94.5 ± 0.1 | **98.7** ± 0.0 | **97.2** ± 0.1 | **99.0** ± 0.0 |
| | ForceAtlas2 | **98.9** ± 0.1 | **97.7** ± 0.0 | **97.6** ± 0.0 | **96.4** ± 0.0 | **94.7** ± 0.0 | 98.6 ± 0.0 | **97.2** ± 0.1 | 98.8 ± 0.1 |
| | $t$-FDP | 97.8 ± 0.0 | 96.1 ± 0.1 | 93.7 ± 0.3 | 96.1 ± 0.1 | 94.3 ± 0.2 | **98.9** ± 0.0 | 88.8 ± 0.5 | 92.8 ± 0.6 |
| | Laplacian E. | 95.7 ± 0.0 | 88.9 ± 0.0 | 91.9 ± 0.0 | 89.8 ± 0.0 | 86.7 ± 0.0 | 96.7 ± 0.0 | 89.7 ± 0.3 | 93.4 ± 1.4 |

Table S5: Spearman correlation between the shortest-path distances and the embedding distances (Euclidean for 2D embeddings, cosine for 128D embeddings). The same setup as in Table S1.

| $d$ | Method | Citeseer | Cora | PubMed | Photo | Computer | MNIST | arXiv | MAG |
|---|---|---|---|---|---|---|---|---|---|
| 128 | **graph NE** | **67.4** ± 1.0 | **81.6** ± 0.1 | **71.4** ± 0.3 | **82.4** ± 0.3 | **75.6** ± 0.0 | **54.2** ± 0.6 | **60.7** ± 0.6 | 7.3 ± 0.8 |
| | **graph NE**$\tau$ | 48.4 ± 0.2 | 62.6 ± 0.3 | 59.2 ± 1.2 | 75.1 ± 0.1 | 70.5 ± 0.1 | 35.0 ± 1.0 | 46.8 ± 2.1 | 12.6 ± 0.3 |
| | CNE, $\tau = 0.5$ | 33.5 ± 0.0 | 35.8 ± 0.1 | 32.9 ± 0.2 | 39.1 ± 0.0 | 32.0 ± 0.0 | 50.8 ± 0.0 | 7.8 ± 0.4 | 21.1 ± 0.3 |
| | DeepWalk | 31.1 ± 0.3 | 25.9 ± 0.6 | 17.0 ± 0.9 | 48.1 ± 0.4 | 34.6 ± 0.8 | 48.3 ± 0.9 | −6.0 ± 0.5 | 9.7 ± 1.8 |
| | node2vec | 27.1 ± 1.3 | 27.5 ± 0.5 | 23.6 ± 0.8 | 47.9 ± 0.3 | 42.7 ± 0.3 | 35.1 ± 1.5 | 17.4 ± 3.0 | 15.5 ± 1.8 |
| | Laplacian E. | 29.6 ± 0.0 | 26.1 ± 0.0 | 37.7 ± 0.0 | 25.8 ± 0.0 | 30.0 ± 0.0 | 26.8 ± 0.0 | 44.8 ± 0.2 | **37.5** ± 0.4 |
| 2 | **graph NE** | 56.9 ± 1.5 | 47.6 ± 3.2 | 35.0 ± 2.6 | 58.3 ± 1.0 | 47.2 ± 2.0 | 41.6 ± 0.3 | 32.9 ± 0.9 | 34.0 ± 2.4 |
| | SGtSNEpi | 46.5 ± 6.1 | 40.0 ± 2.1 | 33.8 ± 1.7 | 52.5 ± 2.7 | 44.1 ± 1.7 | 37.8 ± 5.3 | 33.8 ± 3.0 | 29.9 ± 4.1 |
| | DRGraph | 61.8 ± 3.2 | 65.1 ± 0.9 | 41.3 ± 1.2 | 64.9 ± 1.5 | 49.5 ± 1.9 | 51.3 ± 2.9 | 34.6 ± 1.0 | 31.0 ± 1.0 |
| | ForceAtlas2 | **65.3** ± 0.2 | **71.6** ± 0.1 | 51.0 ± 0.0 | **69.7** ± 0.2 | 54.9 ± 0.0 | **58.2** ± 0.0 | 40.2 ± 0.7 | 23.4 ± 0.5 |
| | $t$-FDP | **65.7** ± 0.1 | 71.1 ± 0.1 | **63.6** ± 0.0 | 64.7 ± 0.4 | **63.8** ± 0.5 | 53.5 ± 0.0 | **56.9** ± 1.7 | **53.6** ± 1.3 |
| | Laplacian E. | 45.5 ± 0.0 | 50.7 ± 0.0 | 21.3 ± 0.0 | 55.4 ± 0.0 | 36.2 ± 0.0 | 52.2 ± 0.0 | 35.7 ± 1.5 | 20.7 ± 0.3 |

Table S6: Top-k ($k = 10$) 2-hop neighbor recall. The same setup as in Table S1. See Appendix C.

| $d$ | Method | Citeseer | Cora | PubMed | Photo | Computer | MNIST | arXiv | MAG |
|---|---|---|---|---|---|---|---|---|---|
| 128 | **graph NE** | **44.5** ± 0.0 | **46.3** ± 0.0 | **35.9** ± 0.0 | 27.2 ± 0.1 | **25.9** ± 0.1 | **57.8** ± 0.1 | **30.3** ± 0.1 | **30.9** ± 0.2 |
| | **graph NE**$\tau$ | **44.3** ± 0.0 | **46.1** ± 0.1 | **35.4** ± 0.1 | 26.7 ± 0.0 | **25.6** ± 0.0 | **57.7** ± 0.1 | **30.2** ± 0.1 | 29.6 ± 0.2 |
| | CNE, $\tau = 0.5$ | 38.0 ± 0.0 | 38.8 ± 0.1 | 30.5 ± 0.1 | 21.4 ± 0.0 | 20.0 ± 0.0 | 37.0 ± 0.0 | 22.3 ± 0.1 | 22.3 ± 0.1 |
| | DeepWalk | 39.7 ± 0.1 | 40.2 ± 0.3 | 32.8 ± 0.3 | 25.4 ± 0.1 | 22.0 ± 0.2 | 53.1 ± 0.6 | 25.9 ± 0.1 | 30.7 ± 0.2 |
| | node2vec | 41.8 ± 0.1 | 43.7 ± 0.1 | 33.4 ± 0.1 | **30.9** ± 0.1 | 24.4 ± 0.1 | 36.1 ± 0.3 | 20.9 ± 0.1 | 14.2 ± 0.1 |
| | Laplacian E. | 36.5 ± 0.0 | 38.0 ± 0.0 | 26.5 ± 0.0 | 23.1 ± 0.0 | 19.8 ± 0.0 | 36.5 ± 0.0 | 20.4 ± 0.0 | 27.0 ± 0.2 |
| 2 | **graph NE** | **35.3** ± 0.1 | **35.0** ± 0.1 | 28.2 ± 0.1 | **24.0** ± 0.1 | 21.8 ± 0.2 | **39.5** ± 0.2 | **22.0** ± 0.0 | **20.2** ± 0.3 |
| | SGtSNEpi | 34.1 ± 0.1 | 33.0 ± 0.4 | **28.7** ± 0.2 | 23.8 ± 0.2 | 21.9 ± 0.1 | 24.6 ± 0.1 | 14.2 ± 0.2 | 5.2 ± 0.1 |
| | DRGraph | 29.2 ± 0.5 | 24.1 ± 0.5 | 11.0 ± 0.2 | 9.8 ± 0.2 | 6.1 ± 0.2 | 7.1 ± 0.1 | 3.8 ± 0.1 | 2.1 ± 0.2 |
| | ForceAtlas2 | 25.3 ± 0.0 | 20.1 ± 0.1 | 14.0 ± 0.1 | 11.3 ± 0.1 | 5.8 ± 0.1 | 4.8 ± 0.0 | 3.2 ± 0.1 | 1.4 ± 0.1 |
| | $t$-FDP | 17.3 ± 0.0 | 13.8 ± 0.2 | 3.8 ± 0.1 | 11.6 ± 0.2 | 5.1 ± 0.1 | 11.5 ± 0.2 | 0.6 ± 0.0 | 0.2 ± 0.0 |
| | Laplacian E. | 18.9 ± 0.0 | 14.7 ± 0.0 | 13.0 ± 0.0 | 8.4 ± 0.0 | 3.5 ± 0.0 | 3.2 ± 0.0 | 1.3 ± 0.2 | 0.5 ± 0.1 |

Table S7: Learned temperature $\tau$ for the graph NE$\tau$ variant in Tables S1–S5. Means across three training runs reported. Standard deviations were alswways below 0.0005.

| $d$ | Method | Citeseer | Cora | PubMed | Photo | Computer | MNIST | arXiv | MAG |
|---|---|---|---|---|---|---|---|---|---|
| 128 | **graph NE**$\tau$ | 0.071 | 0.077 | 0.070 | 0.079 | 0.076 | 0.057 | 0.058 | 0.042 |

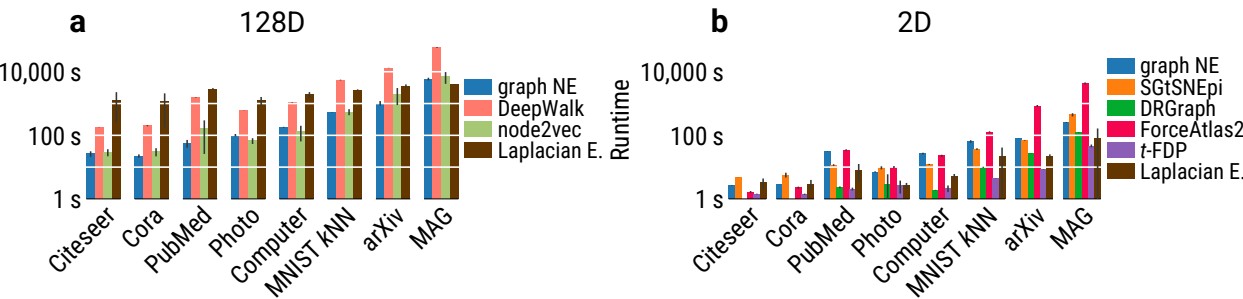

Figure S1: Computation times. All computations were performed on a cluster which isolates the computing resources and removes interference between concurrent computations. All 2D experiments require only CPUs and were ran on 8 cores of an Intel Xeon Gold 6226R. Experiments in 128D ran on a single Nvidia 2080ti GPU card. For node2vec, this shows runtime for $p = q = 1$; we ran 25 parameter combinations (Figure S2), so our actual runtime including hyperparameter tuning was much larger.

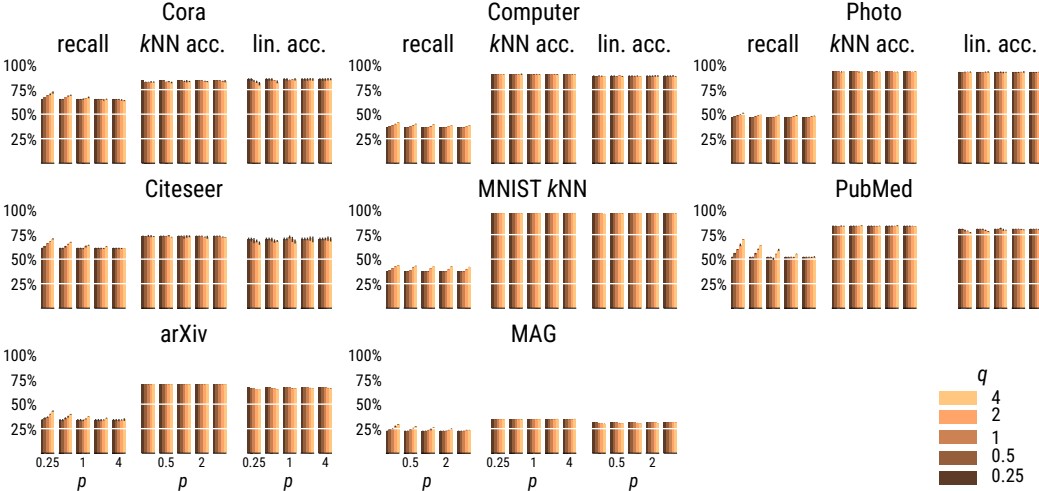

Figure S2: Evaluations for node2vec (Grover & Leskovec, 2016) with the hyperparameter sweep over $p, q \in \{0.25, 0.5, 1, 2, 4\}$. These parameter values were taken from the original node2vec paper. The highest neighbor recall was always achieved at $p = 0.25$, $q = 4$. This corresponds to oversampling unseen nodes.

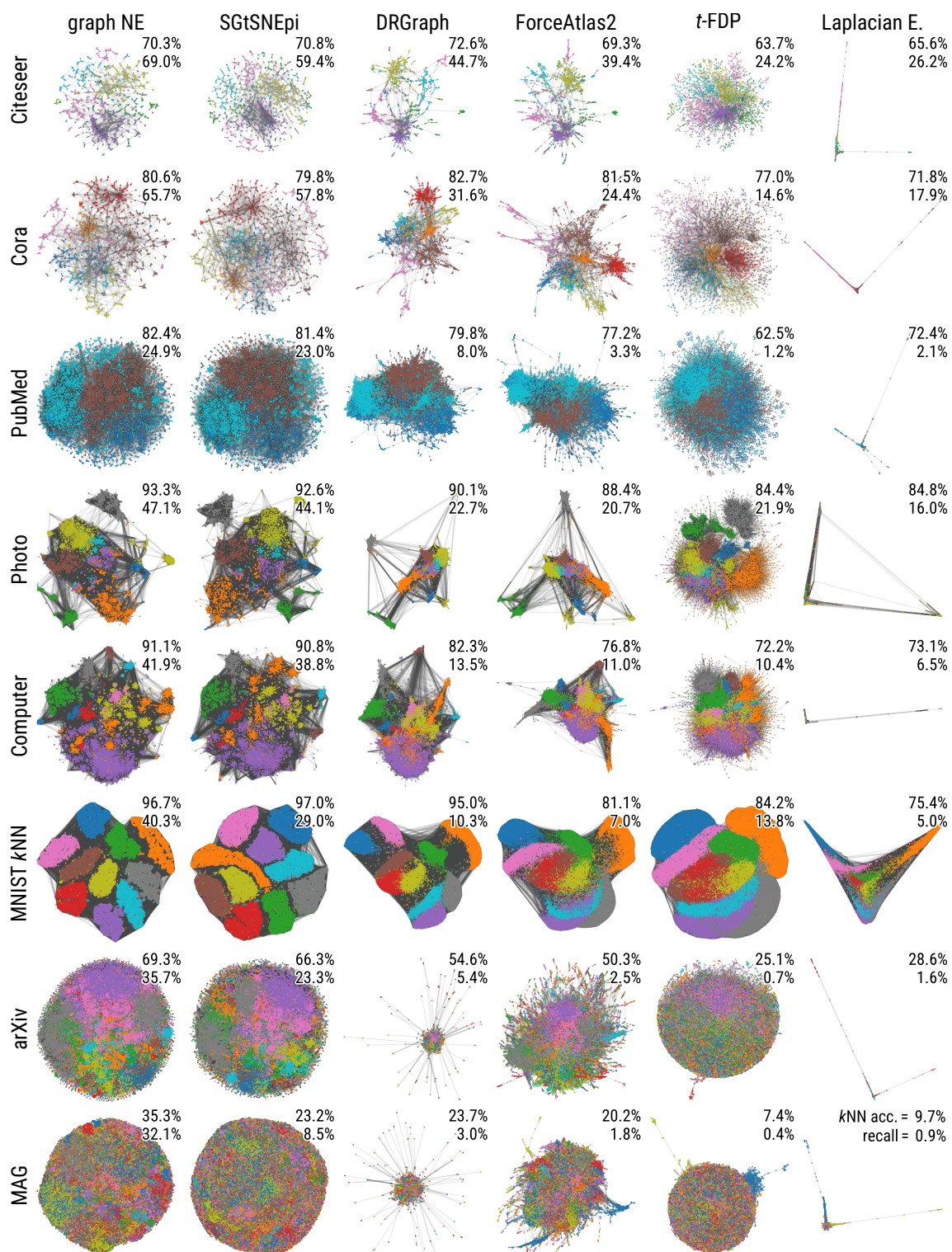

Figure S3: Embeddings of all considered datasets obtained using our graph NE, SGtSNEpi (Pitsianis et al., 2019), DRGraph (Zhu et al., 2020a), ForceAtlas2 (Jacomy et al., 2014), t-FDP (Zhong et al., 2023), and Laplacian Eigenmaps (Belkin & Niyogi, 2003). Embeddings in each row were aligned using orthogonal Procrustes rotation (Schönemann, 1966). Numbers in the top-right corner correspond to the kNN accuracy and the neighbor recall, respectively (as indicated in the bottom-right panel).

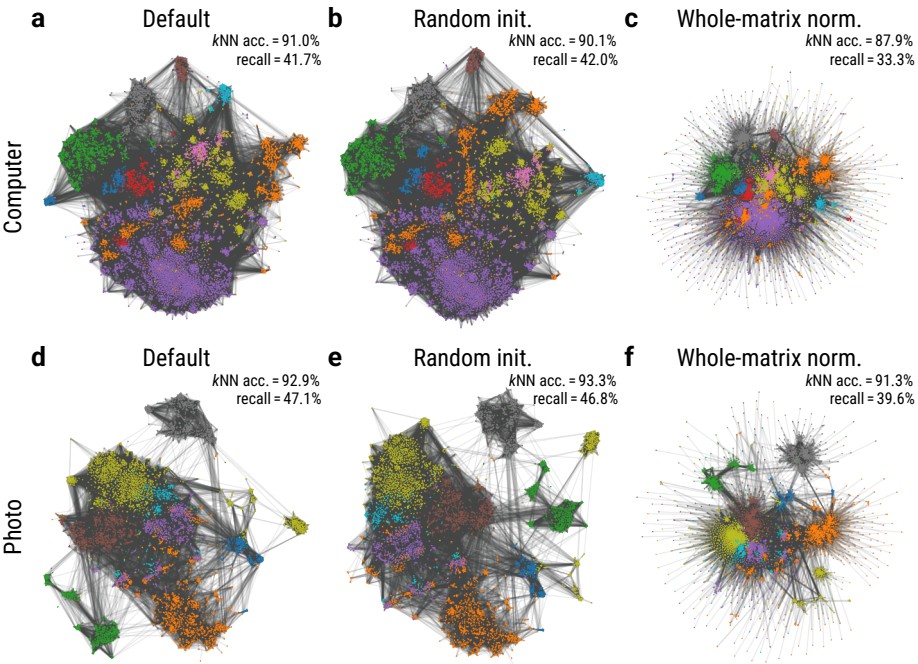

Figure S4: The effect of initialization and normalization on 2D graph NE for the Computer and Photo datasets. **(a, d)** Default graph NE with Diffusion Maps initialization and using per-node normalization of the adjacency matrix. **(b, e)** Graph NE using random initialization. **(c, f)** Graph NE with whole-matrix normalization. Embeddings in each row were aligned using Procrustes rotation.

