# OpenReview forum: "Node Embeddings via Neighbor Embeddings"
_TMLR — Accepted by TMLR_

### Review · Reviewer_AE4y · 2025-09-03

**Summary Of Contributions:**

The paper introduces Graph Neighbour Embeddings (Graph NE), a non-parametric node embedding method that avoids random walks and instead directly optimises embeddings by pulling adjacent nodes closer together. Empirical results on multiple datasets are given, and compared with those of other non-parametric methods like DeepWalk and node2vec. The method is also applied to the problem of 2D graph layouts, comparing its results with classical layout algorithms.

**Additional Comments:**

The caption of Table S4 speaks about Link prediction accuracy, while in the text in Appendix C it is stated that the area under the ROC curve has been computed. The latter is the standard metric and should be reported.

On page 4, it is stated that "the cosine distance and the squared
Euclidean distance differ only by a constant" which is slightly imprecise. It would be better to say "the cosine distance and the squared
Euclidean distance differs only by a constant factor"

**Audience:**

Yes

**Audience Explanation:**

Although the applicability of the paper may be limited (nearly all real-world graphs do have node features) it provides both novel insights (showing that pulling neighbours directly might be effective) and practical impact (potentially better embeddings and graph layouts), which should be of interest to a segment of TMLR’s audience.

**Broader Impact Concerns:**

I do not have any major concerns. The work is mainly methodological, advancing unsupervised graph embedding techniques. It does not directly raise ethical risks or problematic applications beyond the usual caveats for graph representation learning. As with all graph embedding methods, there is a possibility of bias amplification when applied to social or citation networks, but this is not specific to the proposed method

**Claims And Evidence:**

Yes

**Claims Explanation:**

The evidence is mostly convincing, with extensive experiments on multiple datasets and comparisons to several baselines. However, some points would benefit from further clarification and extension:


In particular, I am not convinced by the use of the proposed recall metric to prove the superiority of the approach compared to SoTA solutions.
All direct neighbours of a node are treated equally, which does not seem to be what users usually want when, for example, visualising a network in 2D. Usually, nodes which share more common neighbours should be placed in closer proximity than nodes with no common neighbours.

I suggest defining a distance metric based on this notion (or a normalised version of it) and then evaluating, for example, with precision@k (with k being, for example, 1, 3, 10, ...) as well as the Spearman correlation between this new distance metric and the embedding distances. This way, the shares of the k closest nodes in the embedding are actually among the most similar nodes in the original graph, using a finer metric than just direct neighbourhood (which still could be used as a first discriminator for distance)

Another concern is the performance in terms of computation times in 2D embeddings for many smaller networks (as evidenced in S2b). It seems to be worse than competing algorithms in smaller networks by several orders of magnitude. This may put limits to the algorithms' practical usability in visualisation tools, like for example Gephi, where users tend to expect results within a few seconds.

Furthermore, the authors show that their results may depend on the temperature parameter τ, but only show the effect of two parameter values and leave more detailed experiments to future work. However, in the Appendix, results for NEτ are shown (which seem to optimise for this parameter) but are never explained in the main text, nor are the corresponding values of the temperature parameter τ reported. I suggest that since there are already results reported, the authors should extend the manuscript and at least explain these results, and mention a range of suitable values of the parameter.

Finally, there have been studies (see, for example, [1]  that show how by using also the degrees of the neighbours when encoding a node, performance in link prediction tasks can be improved compared to DeepWalk and node2vec. The authors might want to compare their method against this setting.

[1 ]Alvarez-Gonzalez et al. (2023). Beyond Weisfeiler–Lehman with Local Ego-Network Encodings. Machine Learning and Knowledge Extraction, 5(4), 1234-1265.

An additional advantage of the method used in the study mentioned above is that it is inductive, meaning that it is able to encode new nodes that are not known when the graph is first encoded. The authors might also want to comment on a limitation of their method in this respect (If I understood correctly, it is transductive).

**Requested Changes:**

- Consider incorporating a common-neighbours-based proximity metric (or a normalised variant) in addition to direct adjacency, and evaluate with measures such as precision@k or Spearman correlation.

- Explain better that for smaller networks, the performance in terms of Computation times in 2D embeddings is much worse than competing algorithms.

- Give more details about the NEτ variant of their method and the impact of other values of their temperature parameter

- Explain the limitations of their study regarding inductive graph embedding techniques

- Add computation times to the visualisations provided in Figure 5 and Figures S3

- Report the area under the ROC curve for the link prediction tasks as stated already in the Appendix.

---

> ### Author Response · Authors · 2025-10-17
> **Response to Reviewer AE4y**
>
> Dear Reviewer, many thanks for your constructive review!  We have revised the manuscript and you can find a PDF version of our manuscript with tracked changes as Supplementary Material.
>
>
> > I am not convinced by the use of the proposed recall metric [...] Usually, nodes which share more common neighbours should be placed in closer proximity [...]. I suggest defining a distance metric based on this notion [...]
>
> Following your suggestion, we have now included a new evaluation metric, *Top-k 2-hop neighbor recall* (Table S6). It restricts the normal neighbor recall to the $k=10$ most similar neighbors of each node in terms of the number of shared neighbors (see the new Appendix D). Our graph NE performed very well in this metric. In $d=128$ it was the top performing method on all datasets apart from the Photo dataset, where node2vec achieved a better result. In $d=2$, SGtSNEpi achieved marginally better results on the PubMed and Computer datasets, but Graph NE was the best on all other datasets, sometimes by a large margin (e.g. 15% on the MAG graph).
>
> We only considered a single value of $k$ as this already produced a large table with new results. This additional evaluation underscores Graph NE's strong local structure preservation.
>
> > Explain better that for smaller networks, the performance in terms of Computation times in 2D embeddings is much worse than competing algorithms.
>
> Our graph NE scales linearly in the number of edges of the graph. We therefore expect it to scale very well to large graphs. Most runtime differences to competing methods in terms of embedding optimization are due to implementation details, like the programming language or the precise convergence criterion / number of iterations, and are not `intrinsic' algorithmic differences.
>
> Thanks to your comment, we realized that there was a performance-limiting bug in the openTSNE version we used. The recent commit *a93df4e* fixes this and we now obtain much faster runtimes for the small datasets in 2D (see updated Figure S1). We have added a sentence cautioning about the runtime on small datasets to the Discussion.
>
> In addition, we have now added the computation times for LE to Fig. S1.
>
> > Give more details about the NEτ variant of their method
>
> We described the variant with learned $\tau$ at the end of section 6. We also reported the performance for all three settings (learned $\tau$, $\tau=0.5$, $\tau = 0.05$) in Supplementary Tables S1--S5. We have now cross-referenced to these results in Section 6, mentioned that node embeddings with $\tau=0.5$ are usually worse than with $\tau=0.05$ or learned $\tau$, and added a new Table S7, which gives the final temperature in the learned $\tau$ setting for each dataset.
>
> > there have been studies (see, for example, [Alvarez-Gonzalez et al. 2023]) that show how by using also the degrees of the neighbours when encoding a node, performance in link prediction tasks can be improved compared to DeepWalk and node2vec. The authors might want to compare their method against this setting.
>
> Thank you for pointing us to this interesting reference. Instead of optimizing node embedding coordinates freely, Alvarez-Gonzalez et al. suggest IGEL embeddings, which are DeepWalk with a constrained embedding which depends on the degree structure of the graph. Such an approach is compatible with any gradient-descent-based embedding algorithm, including our graph NE. Since our benchmark focuses on different objective functions rather than constrained embedding spaces, we leave an empirical evaluation for future work. We added a paragraph on Alvarez-Gonzalez et al. to the Related Work section, where we now also explicitly mention the transductive nature of our graph NE.
>
> > Report the area under the ROC curve for the link prediction tasks
>
> Thanks for catching that! We did originally report the area under link-prediction ROC curve in Table S4, and have fixed its caption now.
>
> > It would be better to say "the cosine distance and the squared Euclidean distance differs only by a constant factor"
>
> We made our statement more precise, as suggested, thanks!

---

### Review · Reviewer_H86V · 2025-09-14

**Summary Of Contributions:**

ContributionsThe authors extend the fields of non-parametric node embedding and graph layouts by introducing graph NE. They combine normalization of the adjacency matrix with InfoNCE.

I appreciate the work done by the authors and the extent to which their method has been tested on multiple datasets using different performance metrics, and the many visualisations provided in the Appendix.

**Audience:**

Yes

**Audience Explanation:**

This paper is interesting to TMLR’s audience because it presents a novel way of embedding and visualizing networks. This has implications outside this work as well, since qualitative visualisations can support the results and story of other network-based papers as well.

**Claims And Evidence:**

Yes

**Claims Explanation:**

The authors claim to introduce a non-parametric graph representation learning method. They say that it can be used for both high dimensional embeddings as well as visualisations.
The authors present an extensive experiment testing both the embedding power (in 128D) and give ample visualisations of the result sin 2D.

**Requested Changes:**

•	I believe the authors should elaborate more on the hyperparameters. They are fixed, but I believe this should be justified clearly. This includes the choice to stick to dimension 128.
•	Especially for visualisations, calculation time is the major bottle neck. I think this should get a more prominent role in the paper. Now, it is put away in appendix. Regarding these tests, the authors should clarify why LE is not present in the results of Figure S1.
•	Regarding the graph layouts in 2D in Section 4.3, are the authors applying Equation (7) for graph NE, or SGtSNEpi? Looking at the results from the experiments, it seems like SGtSNEpi is only used as a method to compare to, but including it in Section 4.3 causes some confusion.
•	In Section 4.3, the authors quickly mentioned ‘visually unpleasing embeddings.’ It would be informative if the authors reported why this is unpleasing, and what metrics (if any) they used for this. A visualization being ‘visually pleasing’ could be a qualitative metric the authors could report for Figures S3 and S4 as well.
•	In the introduction, at the end of the second paragraph, ‘neighbor-embedding approaches’ is mentioned for the first time, but it is not entirely clear to what part of the methods mentioned in paragraph it applies. It should be better to also introduce the term in the same way as the authors did with ‘node-embedding methods’ (start of the same paragraph).
•	I believe the last paragraph of the related work part on parametric node embeddings (the part in italics) should be kept for the methodology section.

---

> ### Author Response · Authors · 2025-10-17
> **Response to Reviewer H86V**
>
> Dear Reviewer, many thanks for your encouraging and constructive review.  We have revised the manuscript and you can find a PDF version of our manuscript with tracked changes as Supplementary Material.
>
> > I believe the authors should elaborate more on the hyperparameters
>
> As mentioned in our Sections 4.2 and 4.3, we mostly stick to the hyperparameter defaults of CNE and openTSNE that we use as backends. We described all deviations from the defaults, e.g., for the temperature, in Section 6, and added some further motivation for our choices in Section 4.2. As you noted, we kept the hyperparameters fixed across graphs, unlike in the node2vec experiments, where we performed a grid search over $p$ and $q$ values (Figure S2) to compare against the strongest version of our competitor.
>
> > This includes the choice to stick to dimension 128.
>
> We chose embedding dimension $d=128$ for the high-dimensional node embeddings following the DeepWalk paper. We have now added this motivation to Section 4.2. This high embedding dimension complements the very low-dimensional case of 2D visualization and shows that graph NE works well in both, high-dimensional and low-dimensional embedding spaces.
>
> > Especially for visualisations, calculation time is the major bottle neck. I think this should get a more prominent role in the paper. Now, it is put away in appendix.
>
> Our graph NE scales linearly in the number of edges of the graph. We therefore expect it to scale very well to large graphs. Most runtime differences to competing methods in terms of embedding optimization are due to implementation details, like the programming language, and specific choices of convergence criteria / number of iterations, and are not `intrinsic' algorithmic differences.
>
> We realized that there was a performance-limiting bug in the openTSNE version we used. The recent commit *a93df4e* fixes this and we now obtain much faster runtimes for the small datasets in 2D (see updated Figure S1). We have added a sentence cautioning about the runtime on small datasets in the Discussion.
>
> > the authors should clarify why LE is not present in the results of Figure S1.
>
> We have added Laplacian Eigenmaps to Figure S1. Thanks for spotting that it was missing!
>
> > Regarding the graph layouts in 2D in Section 4.3, are the authors applying Equation (7) for graph NE, or SGtSNEpi?
>
> The paragraph on SGtSNEpi in Section 4.3 is meant to contrast graph NE's direct use of Eq (7) with the more complicated setup of the closely related competitor SGtSNEpi. We have clarified this in the revision. In all our 2D graph NE experiments (save for the ablations in Figure S4) we used Eq (7) and never the setup of SGtSNEpi.
>
> > In Section 4.3, the authors quickly mentioned ‘visually unpleasing embeddings.’
>
> By "visually unpleasing" we simply meant the "hedgehog-shape" of the embeddings in Figure S4 c,f with low-degree nodes pushed out to the periphery of the embedding and thus distracting from the main part of the embedding. We have now rephrased Section 4.3 to avoid the term "visually unpleasing". We did not quantify 'visually pleasing' explicitly, but we did observe that the whole-matrix normalization in Figure S4c,f led to lower kNN accuracy and recall (Section 4.3 and Figure S4).
>
> > In the introduction, at the end of the second paragraph, ‘neighbor-embedding approaches’ is mentioned for the first time, but it is not entirely clear to what part of the methods mentioned in paragraph it applies.
>
> Thank you for catching this, you are right. We have now inserted a sentence into the Introduction that introduces the term *neighbor embeddings* and clarifies to which methods it applies.
>
> > I believe the last paragraph of the related work part on parametric node embeddings (the part in italics) should be kept for the methodology section.
>
> Given the popularity of parametric graph neural networks and also given some feedback we received on prior versions of this manuscript, we feel it is important to clearly state that our work is about non-parametric feature-less graph embeddings to which graph neural networks are not applicable.
>
> As a compromise, we have now de-emphasized this paragraph by removing the *italic* formatting and mentioned the non-parametric nature of graph NE in Section 4.3 as well.

---

### Review · Reviewer_tCp4 · 2025-10-06

**Summary Of Contributions:**

The paper proposes a graph node embedding algorithm that builds on ideas from low-dimensional graph visualization, particularly contrastive neighbor embeddings. A theorem apparently shows an equivalence between the optima of the proposed InfoNCE objective and the well-known tSNE objective for embedding high-dimensional data. Experiments show that Graph NE has high neighbor recall, meaning that neighboring points in the graph are embedded among the nearest neighbors in metric space.

**Audience:**

Yes

**Audience Explanation:**

I suspect people working on network embeddings would be interested, but it's not my area so it's hard to say for sure.

**Broader Impact Concerns:**

no concerns

**Claims And Evidence:**

Yes

**Claims Explanation:**

I'm not very familiar with this area but to my understanding the experiments are convincing.

Given the equivalence of tSNE and infoNCE, would an alternative baseline be to compute embeddings from graph Laplacian and then perform tSNE directly on these embeddings?

Regarding the theory, a major point of confusion was the relationship between the affinities q_{ij} and w_{ij} (introduced without explanation in equation (3)). Are these the same? Similarly, I didn't understand the role of q_theta^* in the statement of theorem 4.1. Please define this term in the theorem statement or elsewhere.

**Requested Changes:**

Please see my questions about about the notation and potential baselines.

---

> ### Author Response · Authors · 2025-10-17
> **Response to Reviewer tCp4**
>
> Dear Reviewer, we thank you for your careful review of our manuscript.  We have revised the manuscript and you can find a PDF version of our manuscript with tracked changes as Supplementary Material.
>
>
> > would an alternative baseline be to compute embeddings from graph Laplacian and then perform tSNE directly on these embeddings?
>
> Thank you for this suggestion. First of all, this is *not* a standard way to construct a 2D graph layout, so does not really qualify as a common `baseline'. Second, as can be seen in Figure 4a and Table 2, Laplacian Eigenmaps in 128D typically has *lower* neighbor recall than graph NE in 2D. Applying t-SNE to the LE output would further decrease the recall, and hence would not be competitive to our graph NE.
>
> We have validated this intuition by running t-SNE (openTSNE with all default parameters) on top of the 128D LE output for the Photo graph. We obtained neighbor recall of 6.1, much lower than 128D LE (39.7) and hence also much lower than our graph NE in 2D (46.9). The numbers were similar for the Computer dataset.
>
> > Regarding the theory, a major point of confusion was the relationship between the affinities q_{ij} and w_{ij} (introduced without explanation in equation (3)). Are these the same?
>
> Both $q_{ij}$ and $w_{ij}$ are affinities that quantify similiarity between embedding points $\mathbf y_i$ and $\mathbf y_j$. The difference is that $q_{ij}$ is normalized to sum to 1, which is necessary for t-SNE's cross-entropy loss function, but results in quadratic computational complexity. The key feature of the contrastive setting is that this costly normalization is not necessary, so that non-normalized affinities $w_{ij}$ can be used instead. We highlight this difference by choosing different letters for the normalized and non-normalized affinities. In the revision, we clarified that $w_{ij}$ serve the same role in Eq (3) as $q_{ij}$ in Eq (1).
>
>
> > Similarly, I didn't understand the role of $q_theta^\*$ in the statement of theorem 4.1. Please define this term in the theorem statement or elsewhere.
>
> Thanks for spotting this typo! Any occurance of $q_\theta$ in Theorem 4 should be a $w(\theta)$. We have rectified this in the revision.
>
> In Theorem 4.1 we want to model a probability distribution $p$ (up to a scaling factor) with a parametric model $w(\theta)$ where $\theta$ are parameters from a parameter set $\Theta$. The parameters $\theta^\*$ in the assumption of Theorem 4.1 achieve this goal: $ w_{\theta^\*} \propto p $, i.e., $w_{\theta^\*}$ equals $p$ up to a constant scaling factor. The theorem then states that the set of such $\theta^\*$'s is precisely the same as the set of minimizers of the InfoNCE loss function and the same as the set of minimizers of the KL divergence.
>
> In the context of learning an embedding, the parameters $\theta$ are simply the coordinates of the embedding points $\theta = \mathbf{y} \in \mathbb{R}^{n\times d}$. We have added this clarification in the revision.

---

> > ### Comment · Reviewer_tCp4 · 2025-10-17
> >
> > Thank you for the clarifications.

---

### Decision · Action_Editor_eoZ3 · 2025-11-03

**Recommendation:** Accept as is

**Additional Comments:**

I suggest the authors make the implementation available in the final version.

**Audience:**

Yes

**Audience Explanation:**

reviewers noted that the method will interest researchers working on graph embeddings and network visualisation within the TMLR community.

**Claims And Evidence:**

Yes

**Claims Explanation:**

The paper claims to introduce a non-parametric graph node embedding method that:
- directly optimizes embeddings by pulling neighboring nodes closer without random walks,
- is effective for both high-dimensional embeddings and 2-D graph layouts,
- provides strong empirical performance in terms of neighbor recall and visualization quality, and
- theoretically connects an InfoNCE objective to the t-SNE objective.

All reviewers agree that the claims are supported by experimentally and clear theoretical justification. The improved manuscript addressed prior concerns. The evidence is convincing. The empirical results are strong and consider multiple datasets and added analyses (e.g., 2-hop neighbor recall). Some theory and notation issues identified by reviewers were clarified in revision.